



# Experimental Validation of Parked Loads for a Floating Vertical Axis Wind Turbine: Wind-Wave Basin Tests

Md Sanower Hossain[1] and D. Todd Griffith[1]

[1]UTD Center for Wind Energy, Department of Mechanical Engineering, University of Texas at Dallas, Richardson, Texas, USA

**Correspondence:** D. Todd Griffith (tgriffith@utdallas.edu)

**Abstract.** Parked loads are a major design load case for vertical axis wind turbines (VAWTs) because of persistent high loads on the rotor when in standstill conditions. This paper examines the aerodynamic parked loads of model-scale floating troposkein VAWTs tested in a wind-wave basin to support development and validation of a parked loads model for floating VAWTs. We analyze the effects of wind speed, and turbine solidity (varying number of blades), and rotor azimuth on parked loads and investigate the impact of different platform conditions (comparing locked (fixed tower base) versus floating cases with and without waves). The experimental results indicate that parked loads (for both locked and floating platform conditions) and amplitude of turbine tilting increases with the wind speed, which is expected. The parked loads also increase with the increase of solidity, however the variation in loads in a revolution decreases for 3 blades versus 2 blades. If only aerodynamic parked loads are considered, the turbine with a floating platform exhibits lower parked loads compared to turbine with a locked platform (fixed base) due to the effect of the tilted condition of the floating platform. Moreover, comparison between floating with wind and wave, and floating with wind only cases show that although both exhibits park loads of similar magnitude, the former exhibits more high frequency variation due to coupled effects of wind, wave, and floating platform dynamics. Additionally, we present a semi-numerical tool for estimating parked loads of VAWTs that we improved and validated to predict the floating parked loads. The analytical model accurately predicted the parked load behavior of VAWTs for the range of effects noted above. The datasets from this experimental work can serve as benchmarks for validating other computational parked load estimating tools.

## 1 Introduction

Vertical axis wind turbines (VAWTs) are gaining attention as a candidate for offshore deployments, especially in deep waters where floating platforms are required. Floating VAWTs have the potential to significantly reduce the cost of energy compared to floating horizontal axis wind turbines (HAWTs) Shelley et al. (2018). Additionally, VAWTs have a lower center of gravity which helps to reduce the overturning moment.

One critical aspect of VAWT operation and deployment is understanding the design loads for parked or standstill conditions. Parked loads refer to the forces exerted on a wind turbine when it is in a stationary or non-operational state, such as during





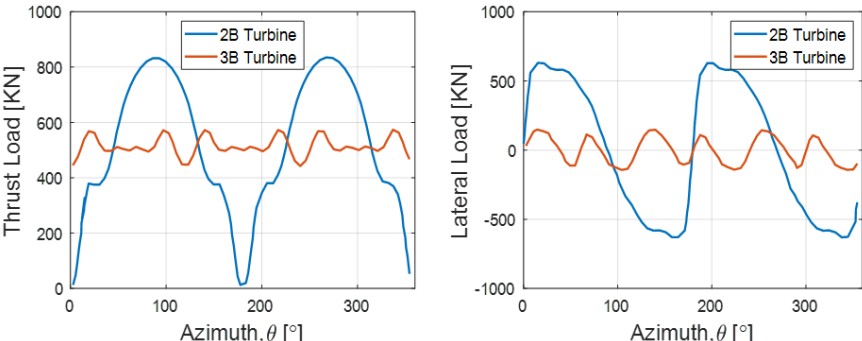

**Figure 1.** Parked loads on 5MW Darrieus VAWT at 30.64 ms$^{-1}$ wind speed [Sakib and Griffith (2022)]

maintenance, low wind conditions, or shutdown scenarios. These loads are influenced by various factors including wind speed,
solidity, number of blades, operating and platform conditions, and the azimuthal position in a revolution.

Accurate assessment of parked loads is essential for several reasons. When not employing a blade pitching mechanism, VAWTs are subjected to high parked loads for extended periods. Sakib et. al. showed that the magnitude of parked loads is similar in magnitude to the operating loads Sakib and Griffith (2022). A sample parked load with respect to azimuth at 30.94 ms$^{-1}$ wind speed for a 5 MW UTD-designed VAWT is shown in Figure 1. Assessing parked loads would ensure the structural
integrity and longevity of the wind turbine and floating system, preventing potential damage that could occur when the turbine is not generating power. Finally, understanding these loads contributes to optimizing the overall design and performance of VAWTs, making them more reliable and efficient.

To date, very few parked load studies of VAWTs are found in the literature. Sakib and Griffith (2022) analyzed parked aerodynamic load for a 5 MW conceptual VAWT, considering rotor design variables such as tapered blade chord, number of
blades, aspect ratio (the ratio of rotor height to diameter). The analytical tool used in Sakib et al's study is validated here and improved by adding the capacity for estimating parked load for offshore floating turbines. Ottermo et al. (2012) developed an analytical model to estimate extreme loads under parked conditions. Paulsen et al. (2013) conducted a CFD study to predict both the operating and parked load for the Deep Wind concept. Kuang et al. (2019) performed a numerical CFD investigation on the flow characteristics and dynamic responses of a parked straight-bladed vertical axis wind turbine (HVAWT) and concluded
that pressure distribution on the upwind blade surface are similar at different azimuthal locations. As wind speed increases, turbulent flow characteristics and wake effects become more pronounced, while dynamic responses due to parked conditions can be neglected. The only experimental parked load analysis of 12 kW VAWT was done by Goude and Rossander (2017), which used fixed base (locked platform) for an HVAWT.

This study focuses on the experimental investigation of parked loads on vertical axis wind turbines with experiments per-
formed in a wind-wave. The research aims to provide a comprehensive understanding of the factors affecting parked loads and their impact on turbine performance. The findings from this study offer valuable insights for the design, operation, and maintenance of VAWTs, ultimately contributing to the advancement of sustainable wind energy technologies.





This paper also focuses on enhancing the capacity of UTD (UT-Dallas)'s existing semi-numerical VAWT parked load estimating tool originally developed by Sakib and Griffith (2022). This tool make use of a mid-fidelity, open-source, free vortex method code CACTUS (Code for Axial and Cross-flow Turbine Simulation) [Murray and Barone (2011)] and analytical methods. CACTUS was developed by Sandia National Laboratories in FORTRAN 95 language using VDART3 [Strickland et al. (1980)] code as a basis. CACTUS code is capable of performing an analysis of any arbitrary turbine configuration by segmenting the turbine blades and struts into a set of blade elements. More detailed explanation of CACTUS can be found in Lu (2020).

In summary, the aim of this work is to experimentally study and validate the modeling of parked loads for Darrieus VAWTs with different platform conditions and to enhance the capacity of UTD's existing VAWT parked load estimating tool. The main contributions of this work can be summarized as below:

- An experimental study is performed that examines troposkein wind turbines with two blades (2B) and three blades (3B).

- The parked dataset presented here is unique because the data includes both fixed base (locked platform, zero tilting), and for floating platform conditions (floating with and without waves).

- Experimental data is gathered to validate an semi-numerical parked load estimating tool, originally developed by Sakib and Griffith (2022). The tool has also been improved to assess the parked load for offshore floating turbines.

Section 2 covers the methods for experiments, the test campaign, and development of the parked loads model. Section 3 presents results and discussions for the experimental measurements and model validation efforts, with a focus on the effects of wind speed, turbine solidity (varying number of blades), different platform conditions, and effect of rotor azimuth. Section 4 presents the concluding remarks.

## 2 Methods

The VAWT rotors were tested at UMaine's Alfond Wind-Wave Ocean Engineering Laboratory (W2) [Cole et al. (2017)]. This unique facility is equipped with high-precision measurement instruments and allows for variable water depths, wind, and wave conditions. The basin measures 30 meters in length and 9 meters in width, with a maximum water depth of 5 meters. The wind machine, whose dimension is 7 m x 3.5 m, can generate up to 12 ms$^{-1}$ wind speed using a narrower condenser passage; however, during these VAWT tests, maximum wind speeds were limited to 4.96 ms$^{-1}$ with a turbulence intensity of 3.9% after removing the condenser. A detailed description of the testing facility can be found in Parker (2022) .

### 2.1 Test turbines

Two scaled turbine configurations were tested at the test facility. These turbines are designed based on the constraints and requirements of the wind wave basin test facility, as well as constraints imposed by the floating system and safety requirements. For example, turbine dimension must not exceed the dimension of wind generating machine (W 7.0 × H 3.5 m). Both the



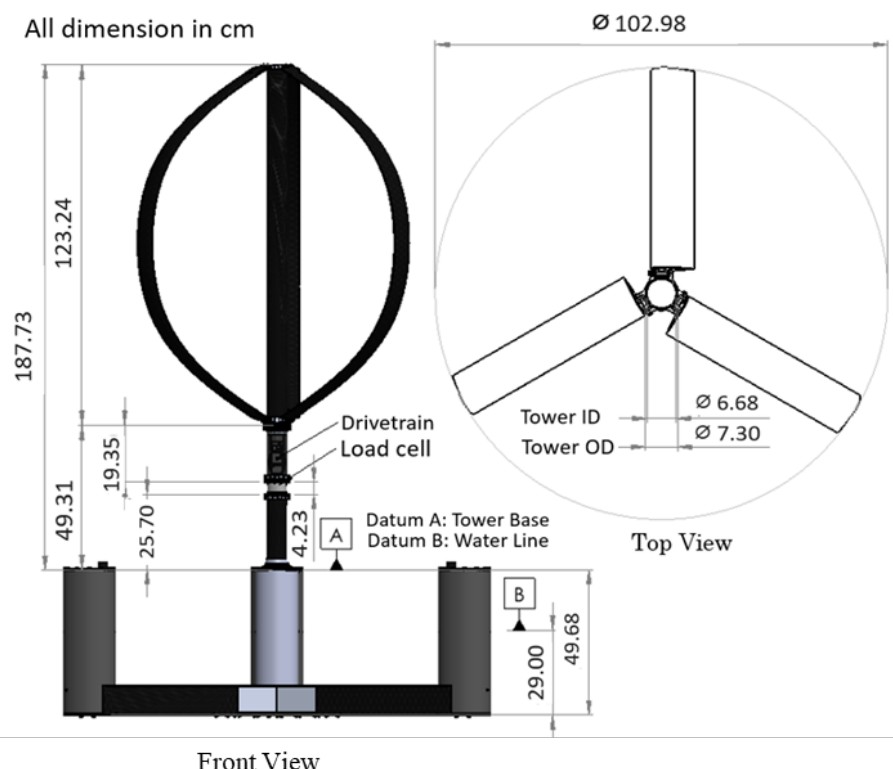

**Figure 2.** Visual of test turbine configuration CAD model with dimension

turbines were alternately mounted on a floating platform which was originally used for FOCAL horizontal axis wind turbine. Details of the floating platform can be found on Robertson (2023). The turbines are also designed in a way that they maintain

the load constraints implemented by the FOCAL platform. The rotor mass and overturning moment must not exceed 12.6 kg and 15.54 Nm, respectively. The overturning moment is defined as the multiplication of thrust force with the distance from mean water level to the center of pressure of the rotor. Moreover, the design met international structural safety requirements from the international design standards of composite wind turbine blades stated in Germanischer (2010). For detailed design of these model scale VAWTs, readers are referred to Hossain et al. (2024).

All the geometric properties for two tested VAWTs are the same except the number of blades. One turbine has two blades, whereas the other has three blades. The shape of the turbine is troposkein with equatorial radius of 0.515 m, and height of 1.287 m. It is configured with NACA0018 airfoil of constant chord length of the airfoil is 0.1 m. The solidity of 2B turbine (two-bladed, 2B) and 3B turbine (three-bladed, 3B) are 0.194 and 0.291, respectively. In this work, the solidity of the turbine is defined as NC/D, where N is number of blades, C is the chord length, and D is the maximum diameter of the turbine. The

solidity varies only due to number of blade variation. The summarized geometric parameters are listed in the Table 1.



**Table 1.** Geometric configurations of two tested troposkein turbines

| Parameter | 2B Turbine | 3B Turbine |
|---|---|---|
| No of blades [-] | 2 | 3 |
| Airfoil chord, [m] | 0.1 | 0.1 |
| Rotor radius, [m] | 0.515 | 0.515 |
| Height to radius, [-] | 2.5 | 2.5 |
| Solidity, NC/D [-] | 0.194 | 0.291 |

**Table 2.** Data measurement system

| Item | Measuring tool |
|---|---|
| Loads and moments | 6 DOF load sensor |
| Rotor speed | Motor encoder |
| Platform pitch/roll | Qualisys 6DOF motion |

A schematic of the test turbine configuration CAD model with dimensions is shown in Figure 2. The image shows both front and top view of a 3B turbine . The dimensions of 2B turbine are same except the number of blades.

## 2.2 Measurement system and data post processing

An ATI Mini58 (SI-700-30 calibration) six degree of freedom (6DOF) load sensor was used to measure the loads and moments.
The load cell can measure maximum thrust of 700 N, maximum overturning moment of 30 Nm, and maximum torque of 30 Nm which meet the expected loads and moment requirements. The load sensor was placed in the tower base, please see Figure 2 for relative position. The rotor speed was measured using Motor encoder. And the platform dynamic motions (pitch/roll) were recorded using a Qualisys 6DOF Motion. Please see Table 2 for data measurements system summary.

The testing facility and turbine design did not include the measurement of blade azimuthal tracking. Therefore, before
every test, blade number 1 was placed at zero-degree azimuth position located at the most upwind position. Then the azimuth variation was tracked running the turbine at a constant and low speed of 1 RPM.

Before starting every measurement, the system data was recorded at non-running/pre-load condition to know if there any residual values exist. After that, the fan was switched on to generate the specified wind speed. Then, after steady-state wind speeds were present, the turbine was rotated at 1 RPM and the raw load data was recorded.





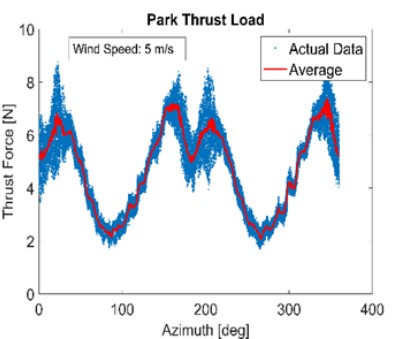 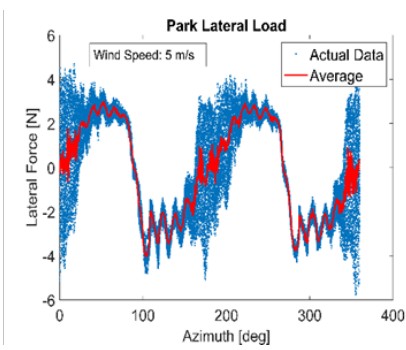

**Figure 3.** Comparison of raw and post processed average thrust and lateral loads at 4.96 ms[-1] for fixed base 2B turbine.

For each test, the data was recorded for at least 5 revolutions. After recording the load data, the raw data was post-processed to exclude the startup (standstill to 1 RPM) and end (1 RPM to standstill) values. After that, the pre-load data was subtracted from the load condition raw data to get the actual true measured data. For averaging the raw data, a binning technique was applied. The azimuthal locations (from 0° to 360°) were divided into 720 bins. Data for all the revolutions are stored in the bins, which indicates that data were comparted in a way that every data for 0.5° azimuthal locations data goes into a single bin. Then the binned data are averaged to get the final post processed data of 720 azimuthal locations.

Figure 3 shows a sample comparison between raw and the post processed averaged parked thrust and lateral loads at 4.96 ms[-1] wind speed for fixed base 2B turbine. The blue points indicate the raw data, and the red continuous line indicates post processed averaged data. This paper made use of the post processed averaged load data for all analysis thereafter.

## 2.3 Test matrix in the test campaign

The test matrix consists of 4 important variables which have the largest impacts on the parked loads of floating VAWTs. The first variable we considered is azimuthal position. The parked loads of VAWTs highly depend on the azimuthal positions. 0° azimuthal positions represents that the first blade is in the windward direction. The turbine rotates anticlockwise. Please see Figure 5 for azimuthal angle convention used in this study.

The second variable we considered is wind speed. The test facility could produce a maximum of 4.96 ms[-1] wind speed. Therefore, the turbines were tested at wind speeds varied between 2 ms[-1] to 4.96 ms[-1] keeping the RPM constant at 1. If we consider the Froude scaled the wind speeds for UTD's 5 MW VAWT [Sakib et al. (2024)] , it results in the range of 14.7 ms[-1] to 36.4 ms[-1]. This wind speed range make quite a sense because the site specific 50 years return period having 10 minutes average wind speed of 30.96 ms[-1] [Sirnivas et al. (2014)] falls in between.

The third variable considered in this test campaign is the number of blades. The two-bladed turbine (2B) has solidity of 0.194, whereas the three-bladed turbine (3B) has solidity of 0.291. The geometric configurations and visual representation of tested turbines are shown in Table 1 and Figure 2, respectively. The detailed design and manufacturing of the turbines can be found in Hossain et al. (2024).



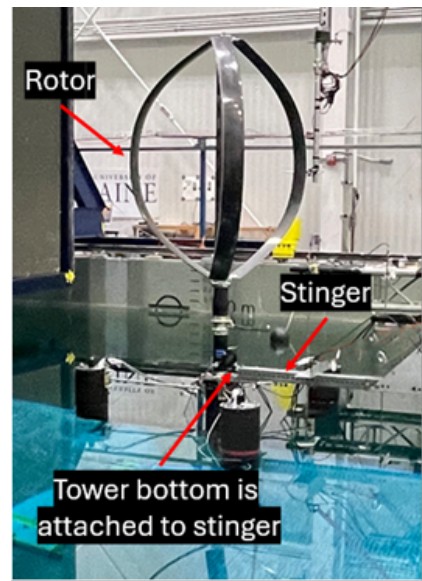 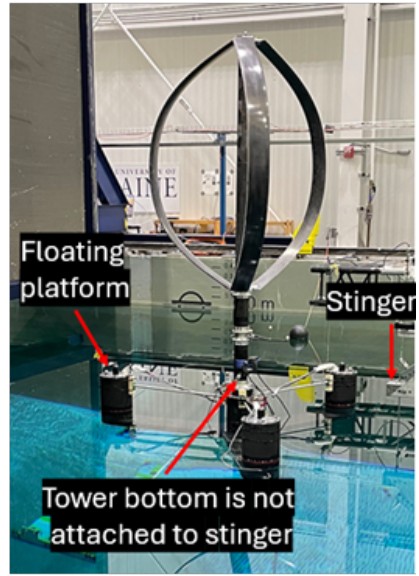

**Figure 4.** Tower base options used in the experiments.

**Table 3.** Operating and Platform Conditions

| Variables | Short Description |
|---|---|
| Azimuth | Azimthal position ranges between 0 ° to 360 °. The position of first blade in the wind ward direction represents the 0 ° azimuthal position. |
| Wind Speed | Wind speeds are varied between 2 to 4.96 ms$^{-1}$. |
| Number of Blades | Two bladed (2B) and three bladed (3B) troposkein VAWTs are tested in this campaign. 2B and 3B turbine has the solidity of 0.194, and 0.291, respectively. |
| Operating Condition | Three operating conditions were considered in the test campaign. Those are locked with wind only, floating with wind only, and floating with wind and wave cases. |

The fourth variable we considered is operating condition. The turbines were tested for three operating conditions: a) locked (fixed tower base) with wind only, b) floating with wind only (no waves), and c) floating with wind and waves. Locked with

wind only condition represents that tower base is attached to stinger to restrict the dynamic movement, and the turbine is rotated only in presence of wind with no waves present. Please see Figure 4 for details of the tower base attached to the stinger. The floating with wind only condition makes use of semi-submersible floating platform. The column of floating platform is



connected to horizontal mooring lines (not shown in the Figure 4) to keep the floating platform in place. However, the mooring lines were not connected to the basin bed. Instead, they were connected to the side wall to reduce the movement for this
specific wind wave basin. In this condition, the turbines are only exposed to the wind. The floating wind and wave condition uses floating platform. However, the turbines are both exposed to wind and waves. In this test campaign, regular waves with 0.155 m height and 1.61 s are used. Readers are referred to Figure 4 for the visuals of locked and floating platforms and referred to Table 3 for test matrix used in this campaign.

## 2.4 UTD's semi-numerical parked load tool

A semi-numerical parked load tool is developed to estimate the parked loads of the tested turbines. As methods like CFD will be very computationally intensive to predict parked loads, a semi-numerical method has been developed with goals of accuracy and low computational effort. This tool makes use of static airfoil polar supplied with the CACTUS tool. Using static airfoil makes sense, as we are estimating the parked load. As one blade does not affect another for parked conditions, no wake effects have been included in the model. Moreover, strut effects, and finite aspect ratio corrections have not been considered.

The local relative velocity and angle of attack for all the azimuthal locations were calculated using CACTUS. In the relative velocity calculation, we only considered local free-stream velocity ignoring the rotational and induced velocity components because at parked condition, rotor is standstill. However, blade wake and tower shadow effects were considered using CACTUS tool.

$$V_N = N_x U_x + N_Y U_Y + N_Z U_Z \tag{1}$$

$$V_N = N_x U_x + N_Y U_Y + N_Z U_Z \tag{2}$$

Where the $V_N$ and $V_T$ are the normal and tangential velocity. $U$ is the freestream velocity. $N$, and $T$ are normal vector component and tangential vector component, respectively. The value of normal vector component ($N$) and the tangential vector component ($T$) are calculated using the 3D vortex-based code CACTUS. The direction of forces and velocities are shown in Figure 5. Where the 0° azimuth is in wind ward direction. The thrust load ($F_{Th}$) is in wind ward direction and the lateral load
($F_{Lat}$) is normal to the thrust load. The normal force ($F_N$) is toward the center and tangential force ($F_T$) is in the tangential direction of airfoil. Local relative velocity is calculated as,

$$V_R = \sqrt{(V_N^2 + V_T^2)} \tag{3}$$

Then, angle of attack ($\alpha$) is calculated from the normal and tangential component of the velocity.

$$\alpha = Tan^{-1}(V_N/V_T) \tag{4}$$

Then the lift ($C_L$) and drag ($C_D$) coefficients for respective angle of attack are calculated using the static airfoil polar supplied with CACTUS tool. After that, the local normal ($C_N$) and tangential ($C_T$) force coefficients are calculated using $C_L$ and $C_D$





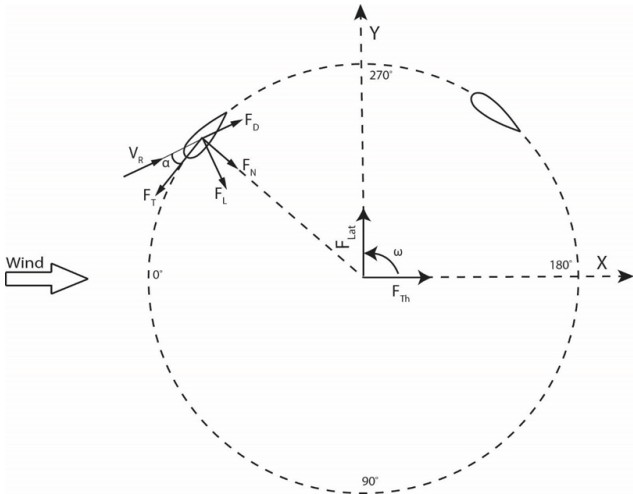

**Figure 5.** VAWT forces and velocities

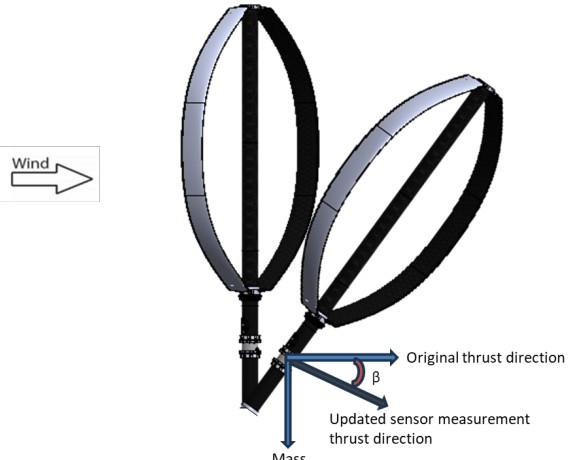

**Figure 6.** Inertial effect on floating VAWT.

as follows.

$$C_N = C_L \, Cos(\alpha) + C_D \, Sin(\alpha) \tag{5}$$

$$C_T = C_L \, Sin(\alpha) - C_D \, Cos(\alpha) \tag{6}$$

Where $C_L$ represents lift coefficient, and $C_D$ represents drag coefficient. The normal and tangential force coefficients are local, in other words with respect to the specific element. Therefore, normal and tangential force coefficients need to be re-referenced





to full turbine scale to calculate the thrust and lateral forces. The conversion is done using the following equations.

$$C_{FtN} = C_N \left(A_E/A\right) \left(V_R/U\right)^2 \tag{7}$$

$$C_{FtT} = C_T \left(A_E/A\right) \left(V_R/U\right)^2 \tag{8}$$

Where, $A_E$ is element area, $A$ is rotor area, $C_{FtN}$ and $C_{FtT}$ are normal and tangential force coefficients w.r.t. rotor, respectively. After that thrust ($C_{Th}$) and lateral ($C_{Lat}$) force coefficients are calculated applying specific normal and tangential direction in the azimuthal locations.

$$C_{Th} = N_x\, C_{FtN} + T_x\, C_{FtT} \tag{9}$$

$$C_{Lat} = N_Y\, C_{FtN} + T_Y\, C_{FtT} \tag{10}$$

Thrust and lateral loads for locked platform case,

$$F_{Th} = 1/2\,\rho A\, C_{Th}\, U^2 \tag{11}$$

$$F_{Lat} = 1/2\,\rho A\, C_{Lat}\, U^2 \tag{12}$$

More detailed description of semi-numerical park load tool for fixed base VAWTs can be found in Sakib and Griffith (2022). The model also captures the tower drag. Tower drag is estimated using Equation 13.

$$F_{DTower} = 1/2\,\rho\, A_T\, C_D\, U^2 \tag{13}$$

Where, $A_T$ represents the frontal tower area, and $C_D$ is drag coefficient. For this analysis, a tapered cylindrical tower with an assumed $C_D$ of 1 was used. The measured thrust load for the locked platform case includes tower drag, while the lateral load does not account for it, as static tower drag force acts in the thrust load direction.

$$F_{ThM} = F_{DTower} + F_{Th} \tag{14}$$

The parked load estimating tool presented in Sakib and Griffith (2022) could only calculate the parked load for fixed base turbine. However, estimating park loads for floating offshore VAWTs is also important. The procedure of estimating park loads for floating offshore VAWTs is outlined below.

The park loads for floating offshore VAWTs are directly correlated with the turbine dynamic motions. For example, updated sensor measurement thrust, and lateral loads are directly correlated with turbine pitch and roll motions, respectively. The
detailed correlation of loads and turbine dynamic motions is shown in section 3.1.4.

The load cell is fixed on the tower base, please see Figure 1 for the load cell position. For the floating platform condition, the turbine tilts due to the coupled effects of wind load and floating system. Therefore, the load cell, which is fixed in the tower, also tilts the same as the turbine. Therefore, the load cell's measured loads will not be in the original thrust or lateral load direction. The measured park loads for are in the updated sensor measurement load direction. Readers are referred to Figure 6
to see the updated sensor measurement load direction. The figure only shows the updated sensor measurement thrust direction.





The original thrust direction is in windward direction. However, due to wind and wave loads the load sensor also tilts. There the measured data corresponds to tilted updated sensor measurement direction. However, being consistent with the locked and floating platform terminologies, we will still call the floating parked loads as thrust and lateral loads.

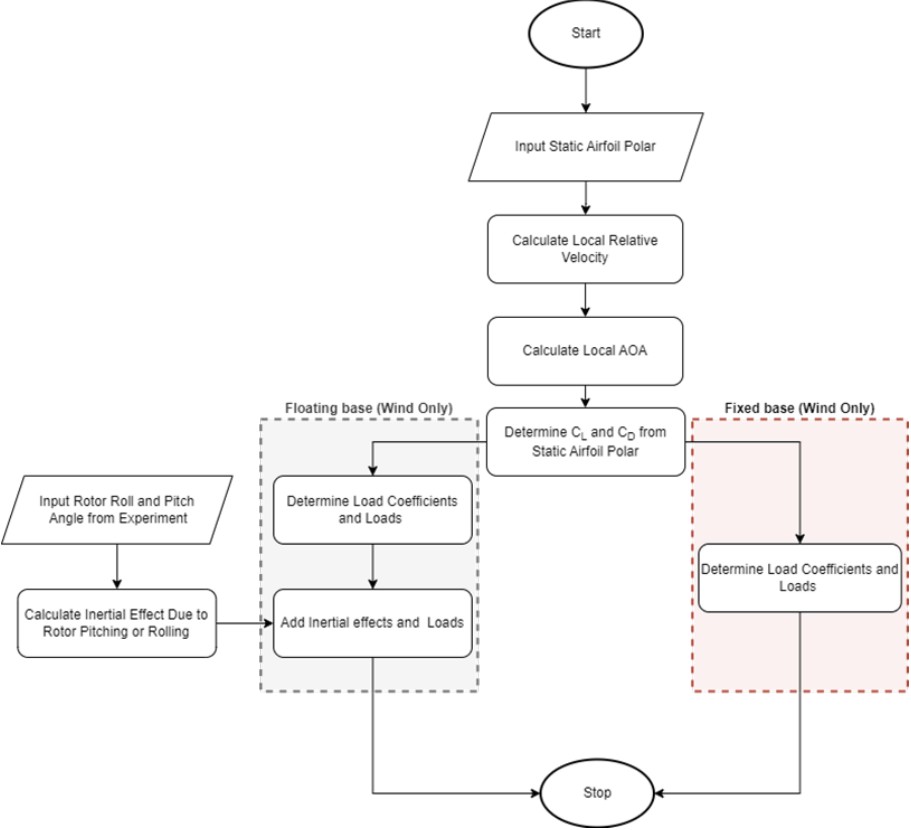

**Figure 7.** Flowchart of parked load estimating semi-numerical tool.

The semi-numerical parked load measuring tool takes this tilting effect into consideration by including the resulting inertial
(weight) effect due to tilting. Due to tilting, the load sensor measures a component of wight (which we are calling here as inertial effect), along with the loads (thrust/lateral) in the original direction. Figure 6 shows the concept of inertial effect present in the floating thrust load. In the test campaign, both the loads and turbine dynamic motions were recorded. We used the measured pitch and roll data to calculate the inertial effect.

$$Weight\,(due\,to\,pitch) = M\,g\,sin(\beta) \tag{15}$$

$$Weight\,(due\,to\,roll) = M\,g\,sin(\gamma) \tag{16}$$





Thrust and lateral loads including both aerodynamic loads and inertial effects for the floating platform case are given in the mode as follows:

$$F_{ThFl} = F_{ThM} + M\,g\,sin(\beta) \tag{17}$$

$$F_{LatFl} = F_{Lat} + M\,g\,sin(\gamma) \tag{18}$$

where $\beta$ is measured pitch angle, $\gamma$ is measured roll angle, $M$ is turbine mass, and $g$ is the gravitational acceleration constant.

A flowchart of the semi-numerical parked load calculation process stated above is also shown in Figure 7. This figure contains park load modeling of both the locked base and floating base platforms. The floating base case compensates for the tilting effect taking inertial effects into consideration.

The UTD's parked load tool for the fixed tower base case has already been validated in Sakib and Griffith (2022). The
extension of the tool for floating tower base case is discussed and validated in the results and discussion section of this paper.

## 3 Results and Discussion

### 3.1 Experimental data analysis

The main objective of this study is to quantify and model the effects of number of blades, wind speeds, operating conditions, and rotor azimuth on parked loads. This section will show and discuss the effects of these conditions successively.

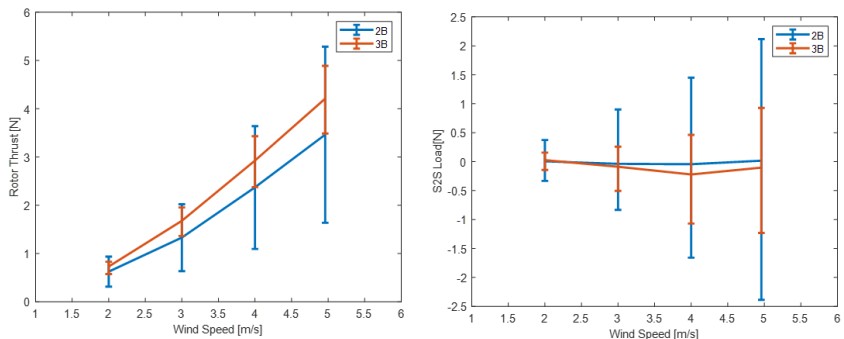

**Figure 8.** Effect of number of blades at locked tower base condition: a. Thrust force b. Lateral Force.

### 3.1.1 Effect of number of blades variation

Both 2B and 3B floating VAWTs were tested in this test campaign. Variations in number of blades, impacts the aerodynamic characteristics, structural dynamics, and overall efficiency of the turbine. For floating VAWTs, understanding the effect of number of blades on parked loads is essential to ensure stability and integrity under various operational and environmental conditions. This section explores the influence of number of blades on parked loads, and their implications for floating VAWTs.




Both 2B and 3B VAWTs were tested for fixed tower base and floating tower base conditions. However, the result is shown for locked wind only case due to the fact that locked and floating platform condition show similar trend.

Figure 8 shows the parked loads as a function of wind speed for both the turbines. It can be seen from the figure that average thrust load increases with the increase in number of blades due to higher solidity and higher blockage to the incoming flow stated in Rezaeiha et al. (2018). Whereas, the number of blades effects are highly azimuth dependent. The variation (range) of

thrust load due to azimuthal dependency in a particular wind speed decreases as the number of blades increases. The average lateral load is independent of number of blades. It always stays at zero as wind has almost zero contribution in the lateral direction. However, the amplitude of lateral forces (load fluctuation for a particular wind speed due to azimuthal dependency) decreases as the number of blades increases which might be helpful for turbine structural integrity Le Fouest and Mulleners (2022).

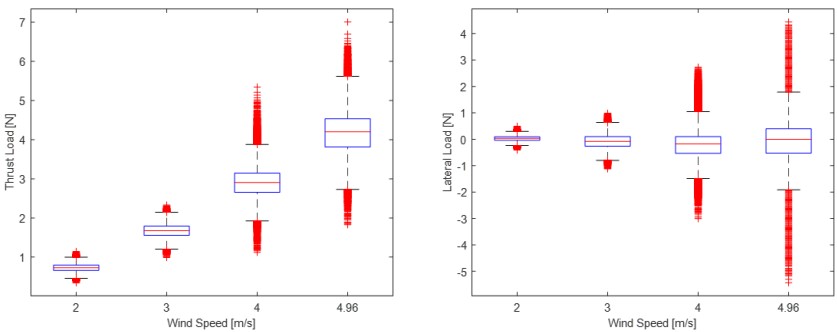

**Figure 9.** Effect of wind speeds on 3B turbine at locked tower base condition: a. Thrust force b. Lateral Force

### 235 3.1.2 Effect of wind speeds

The parked loads of vertical axis wind turbines (VAWTs) are significantly influenced by wind speed. This section investigates the influence of wind speeds on the parked loads of VAWTs. To assess the impact of wind speeds on parked loads, a series of experiments were conducted using a scaled model of a floating VAWT. The turbines were exposed to wind speeds ranging from 2 to 4.96 ms$^{-1}$ in a controlled wind tunnel environment. To keep the paper concise, this section only considers the parked

loads on locked wind only condition. We skipped the effect of wind speed for two bladed turbine due to the fact that the two bladed turbine and three bladed turbine show similar trend on parked loads. The impact of wind speeds on parked loads for three bladed turbine through whisker box plot is shown in Figure 9. The figure shows the data for locked wind only condition. The result indicates that the amplitude of lateral load and average thrust increase with the wind speed. The red dots indicate the outliers. The high number of outlier data is due to the variation of load with respect to the azimuth.

Current vertical axis wind turbine concepts usually exclude the complex pitching of blades to shade some load in higher wind speeds. Therefore, the loads increase with the wind speed is expected as no pitching of blade was present in this study. The increase of parked loads with wind speeds can also be substantiated from the similar comments made in Carmo et al.





**Table 4.** Operating and Platform Conditions

| Conditions | Short Description |
| --- | --- |
| Locked with wind only | Tower base/floating platform is connected to the stinger to restrict the tower base movement. The turbine is rotated in the presence of only wind. |
| Floating with wind only | The floating platform is connected with horizontal mooring line. And the turbine is rotated in presence of only wind. |
| Floating with wind wave | The floating platform is connected with horizontal mooring line. And the turbine is rotated in presence of both wind and wave. |

(2024). This increase of parked load with wind speed signifies that parked load will be a crucial factor in high wind speed. Thus, wind turbine designers should focus on parked loads too, especially in the high wind speed.

### 3.1.3 Comparisons among different operating conditions

In this section, we compare the parked loads among the three conditions detailed in Table 4. In locked (fixed tower base) wind only condition, the tower base is attached to stinger to restrict all the tower base motion. In floating wind only condition, the floating platform is connected to the mooring line. The mooring lines are horizontal line which are connected to the bottom of the floating platform columns and springs. The horizontal mooring was adopted to reduce the experimental uncertainty related to conventional catenary mooring [Ahsan et al. (2022)]. In the floating wind wave condition, the rotor was operated in floating platform in presence of both wind and wave.

In the test campaign, we measured the parked loads for 2, 3, 4, and 4.96 ms$^{-1}$ for locked wind only and floating wind only conditions. Tests were conducted for both two and three bladed turbines. However, parked loads at only 4 and 4.96 wind speeds were measured for floating wind wave condition. Moreover, this case considered 3 bladed turbines only, due to limitations of available testing time. The loads show similar characteristics for two and three bladed turbines.

Offshore floating conditions often cause turbines to tilt, which has two competing effects: greater tilt reduces aerodynamic loads but increases weight effects. Experimental lateral load and thrust load comparison among different conditions are shown on Figure 10 and Figure 11, respectively The results shows that lateral load amplitude and average thrust load for floating conditions is greater than the locked wind only condition. . Although tilting in the floating case can reduce aerodynamic loads due to reduction in swept area and through damping effects [Ahsan et al. (2022)], the added weight component in the sensor's direction results in higher measured loads. When the turbine tilts, the six degree of freedom load and moment measuring sensor also tilts. This sensor tilting along with inertial load measurement is shown in Figure 6. Thus, the measured data includes the lateral load component and inertial load of turbine due to rotor tilting. If we would consider the loads in the original thrust or




lateral direction only, then floating with wind only and floating with wind wave cases would show the expected lower parked
loads compared to locked wind only case.

The lateral load amplitude and average thrust load for floating wind only and floating wind wave are almost the same (please
refer to Figure 10 and 11. The only variation is that the floating wind wave data are noisier (or varied with high frequency) than
floating wind only condition due to the coupled dynamics of wind and wave. The high frequency variation is due to the high
frequency (0.62 Hz) of regular wave used in the experiment.

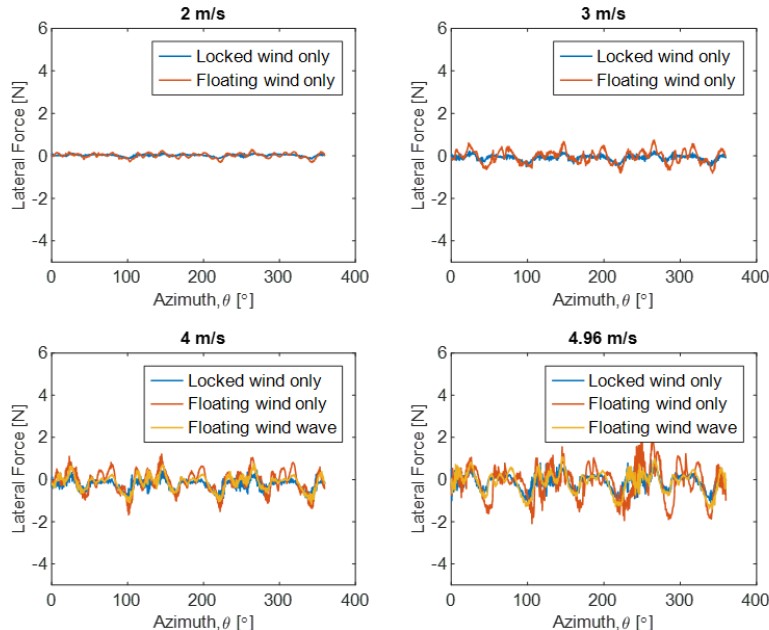

**Figure 10.** Experimental lateral load as a function of azimuth (3B Turbine) a. 2 ms$^{-1}$ b. 3 ms$^{-1}$ c. 4 ms$^{-1}$ d. 4.96 ms$^{-1}$.

### 3.1.4   Correlation between parked loads and turbine dynamics

The correlation coefficient of two random variables is a measure of their linear dependence. If each variable has N scalar
observations, then the Pearson correlation coefficient is defined as

$$P(A, B) = \sum_{i=1}^{N} (A_i - \mu_A \sigma_A)(B_i - \mu_B \sigma_B) \tag{19}$$

where $\mu_A$ and $\sigma_A$ are the mean and standard deviation of $A$, respectively, and $\mu_B$ and $\sigma_B$ are the mean and standard deviation
of $B$.

The loads and dynamic motions with respect to azimuthal position are shown in Figure 12 and Figure 13 for the lateral/roll
and thrust/pitch loads/motions, respectively. The left Y axis represents the load and right Y axis represents the tilt angle. The





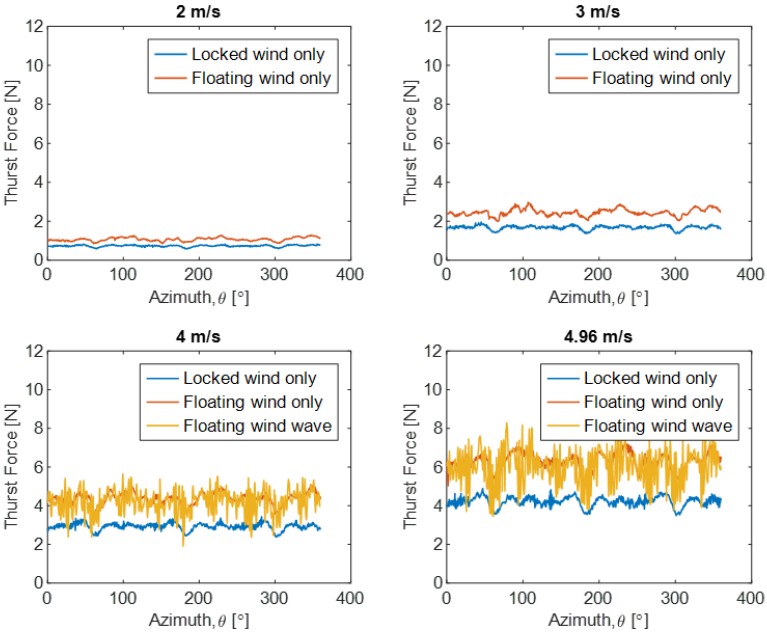

**Figure 11.** Experimental thrust load as a function of azimuth (3B Turbine) a. 2 ms$^{-1}$ b. 3 ms$^{-1}$ c. 4 ms$^{-1}$ d. 4.96 ms$^{-1}$.

correlation coefficients between lateral load and roll motion for 2, 3, 4, and 4.96 ms$^{-1}$ are -0.98, -1, -0.99, and -1, respectively. Whereas the correlation coefficients between thrust load and pitch motion for 2, 3, 4, and 4.96 ms$^{-1}$ are 0.98, 0.99, 0.99, and 285   0.99, respectively. The coefficients shown here are for 2B tubine. It is seen that the dynamic motions and parked loads are highly correlated. Thrust load is highly correlated with pitch motion, whereas lateral load is highly correlated with roll motion. For the lateral loads/roll motion case, the negative correlation coefficient is simply due to the assumed lateral load and roll direction variation. This does not mean that as the roll angle increases, the turbine lateral load decreases.

### 3.2  Semi-numerical parked loads model and validation

### 3.2.1  Locked wind only condition

This section presents parked loads from both the experiments and the semi-numerical model of two-bladed (2B) and three-bladed (3B) turbines with fixed tower base.

Figures 14 and 15 illustrate the variation of parked thrust and lateral loads with respect to azimuthal position for two-bladed turbine and three-bladed turbines, respectively. Analysis shows generally a very good agreement between experiments and 295   our parked loads model in terms of both magnitudes and rotor azimuth dependence. Both thrust and lateral loads amplitude increase with the speed increment. The sharp decrease of experimental thrust load at 180° azimuthal location is due to the tower shadow effect. The model also well captures tower shadow effect.



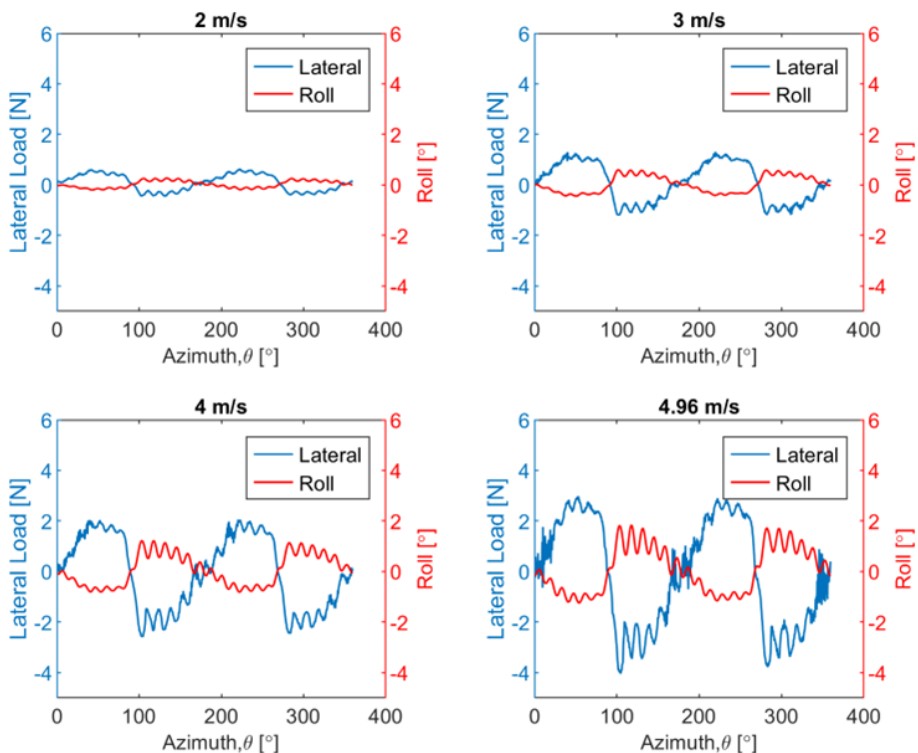

**Figure 12.** Experimental lateral load and dynamic roll motions in floating wind only condition for 2B turbine a. 2 ms$^{-1}$ b. 3 ms$^{-1}$ c. 4 ms$^{-1}$ d. 4.96 ms$^{-1}$.

Both experimental and semi-numerical results show a clear relationship between wind speed and thrust force: higher wind speeds generate larger thrust forces. The periodic nature of the thrust force with respect to azimuth angle is consistent across both methods.

The lateral force increases in magnitude with higher wind speeds, especially at 4.96 ms$^{-1}$. Experimental results show measurement noise due to sensor inaccuracies and inherent variability in the system..The semi-numerical appears smoother compared to measured data. Again, the overall trends of force variation are captured by both the experiment and the semi-numerical model.

Thrust force patterns (Figures 14 and 15) are quite similar in experiment and model, though the experimental results again include measurement noise. Both model and experiment capture the cyclic nature of the forces with respect to the azimuth angle, but the semi-numerical results tend to slightly overpredict the forces at higher wind speeds.

Figures 14 and 15 illustrate the variation of parked thrust and lateral loads with respect to azimuthal position for two-bladed turbine and three-bladed turbines, respectively. Analysis shows generally a very good agreement between experiments and our parked loads model in terms of both magnitudes and rotor azimuth dependence. Both thrust and lateral loads amplitude

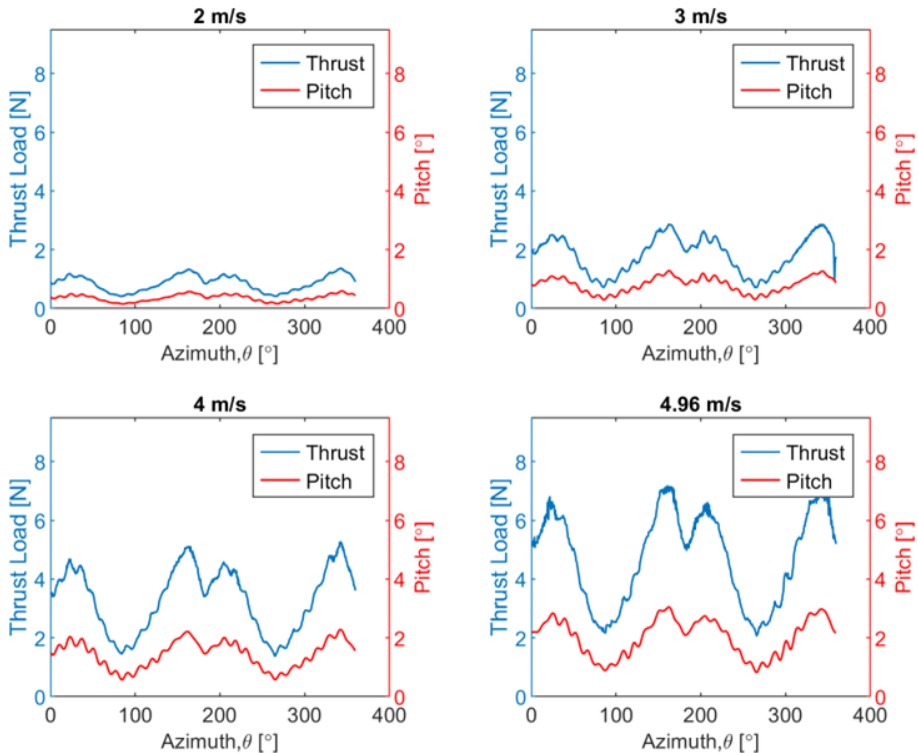

**Figure 13.** Experimental thrust and dynamic pitch motions in floating wind only condition for 2B turbine a. 2 ms[-1] b. 3 ms[-1] c. 4 ms[-1] d. 4.96 ms[-1].

increase with the speed increment. The sharp decrease of experimental thrust load at 180° azimuthal location is due to the tower shadow effect. The model also well captures tower shadow effect.

Both experimental and semi-numerical results show a clear relationship between wind speed and thrust force: higher wind speeds generate larger thrust forces. The periodic nature of the thrust force with respect to azimuth angle is consistent across both methods.

315 The lateral force increases in magnitude with higher wind speeds, especially at 4.96 ms[-1]. Experimental results show measurement noise due to sensor inaccuracies and inherent variability in the system..The semi-numerical appears smoother compared to measured data. Again, the overall trends of force variation are captured by both the experiment and the semi-numerical model.

320 Thrust force patterns (Figures 14 and 15) are quite similar in experiment and model, though the experimental results again include measurement noise. Both model and experiment capture the cyclic nature of the forces with respect to the azimuth angle, but the semi-numerical results tend to slightly overpredict the forces at higher wind speeds.

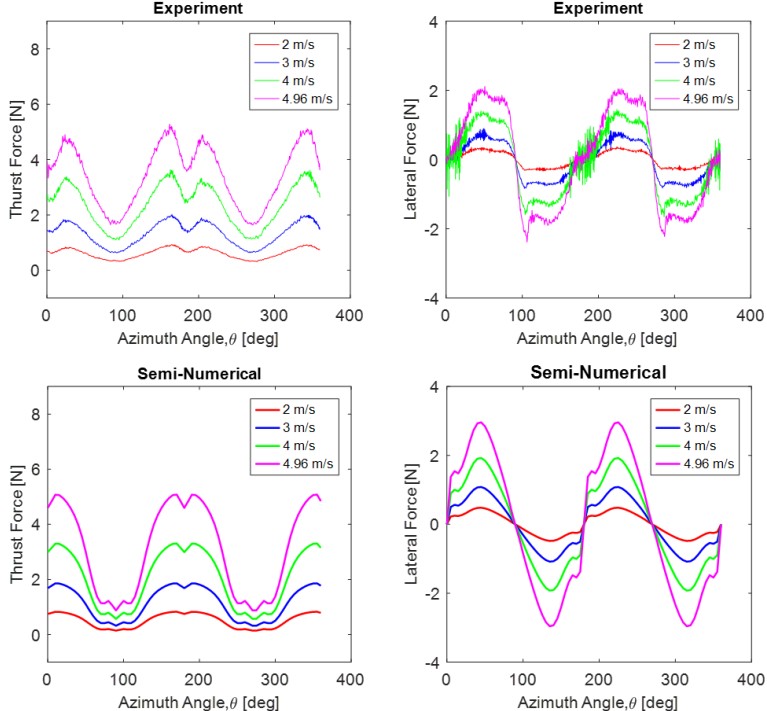

**Figure 14.** Validation of UTD semi-numerical parked load estimating tool for locked with wind only two bladed turbine.

### 3.2.2 Floating wind only condition

Modeling the floating case with wind comes with a new challenge. The challenge is to model the inertial effect due to rotor
tilting in the floating tower base case, as well as capturing the impact of tilting on aerodynamic loads. A detailed description
of modeling parked load at floating tower base condition has been shown in section 2.

This section is intended to analyze the modeled parked load data for floating wind only case in the absence of waves. Due
to floating condition, the rotor tilts under wind loading. Which adds a component of rotor weight in the measured updated
sensor thrust and lateral direction. The rotor weight was already measured in the wind wave basin. The two-bladed and three-
bladed turbines weigh 5.97 kg and 6.55 kg, respectively. This mass includes only the components mounted above the load cell
including the blades and tower mass, excluding the mass of lower tower section, load cell, generator and the hull.

A representative inertial effect in the thrust direction for the two-bladed turbine in the floating wind only condition is shown
in Figure 16. This figure shows the pitch in red and inertial force in the measured thrust direction in blue. It can be seen that,
with the increase in wind speed, the inertial effect also increases. It is a significant amount compared to the parked thrust load
of similar wind speed. If we consider the maximum parked thrust force, the locked wind only case at 4.96 ms$^{-1}$ wind speed
exhibits the value around 5 N. Whereas, the inertial effect at the floating condition due to tilting at 4.96 ms$^{-1}$ is 2.1 N, which is



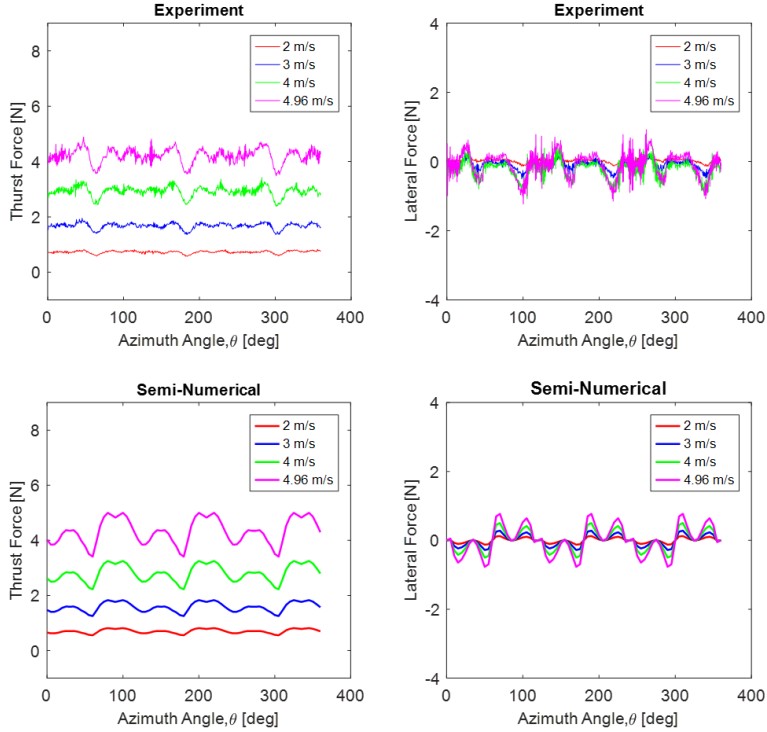

**Figure 15.** Validation of UTD semi-numerical parked load estimating tool for locked with wind only three bladed turbine

around 40% of the thrust load of locked wind only case. Thus including the rotor weight effect is very important at this model scale.

Figures 17 and 18 show the validation of UTD semi-numerical parked load with floating wind only condition for two-bladed and three-bladed turbines, respectively. The tool accurately predicts the trend for both thrust and lateral parked loads for floating wind only case. However, model slightly over predicts the thrust load and under predict the lateral loads. The high frequency variation in loads is due to turbine dynamics (roll/pitch) in the floating condition.

### 3.2.3 Floating tower base with wind and wave

We now examine the third configuration of the test, which is floating with both wind and waves. The inertial effect is also added with the fixed tower base parked load to calculate the parked load for wind and wave case. Only regular waves were used to measure the parked load at floating tower base with wind and wave base. The interesting fact is that the turbine faces both wind and wave. And as the base is floating, it reflects a coupled effects due to wind, wave and floating base.

The validation in the case of floating with wind and wave case for the three-bladed turbine is shown in Figure 19. It shows that the tool neglects the dynamic nature due to floating wind and wave effect. The ongoing work is modeling the coupled wind wave floating dynamics. However, the tool can well predict the average parked lateral and thrust loads for this floating case with



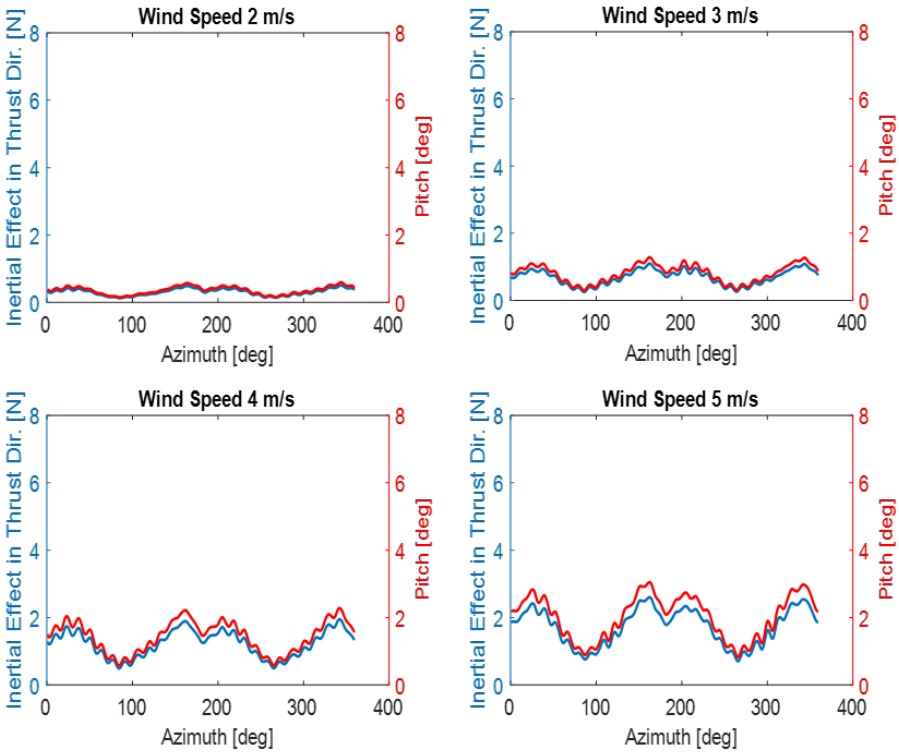

**Figure 16.** Inertial effect in the thrust direction for 2B turbine in the floating with wind only tower base condition

combined wind and wave loading. Overall, the semi-numerical tool well predicts the load magnitude, azimuth dependence, wind speed, and solidity effects for all the operating conditions. This tool can also capture the tower shadow in the 180° azimuthal location.

## 4 Conclusions

Floating offshore VAWTs are showing promise for deep water offshore locations as they offer several advantages including of a lower center of gravity, thus improving stability and reducing the risk of overturning. However, some aspects in the design of floating VAWTs must be studied, including parked loads, which are comparable in magnitude to operating loads [Sakib and Griffith (2022)], are thus critical design loads, yet no studies have measured parked loads under floating conditions.

This study experimentally investigates parked loads on floating VAWTs in a wind-wave basin. The study aims to provide insights on the factors influencing parked loads. The study also aims to gather data to improve and validate a semi-numerical parked load estimation tool for floating VAWTs under both wind-only and wind-and-wave conditions that is based on a vortex aerodynamics model of the rotor and analytical model of tower drag. Further, this model is shown to capture the effects of tower shadow, azimuth dependence and inertial effects.



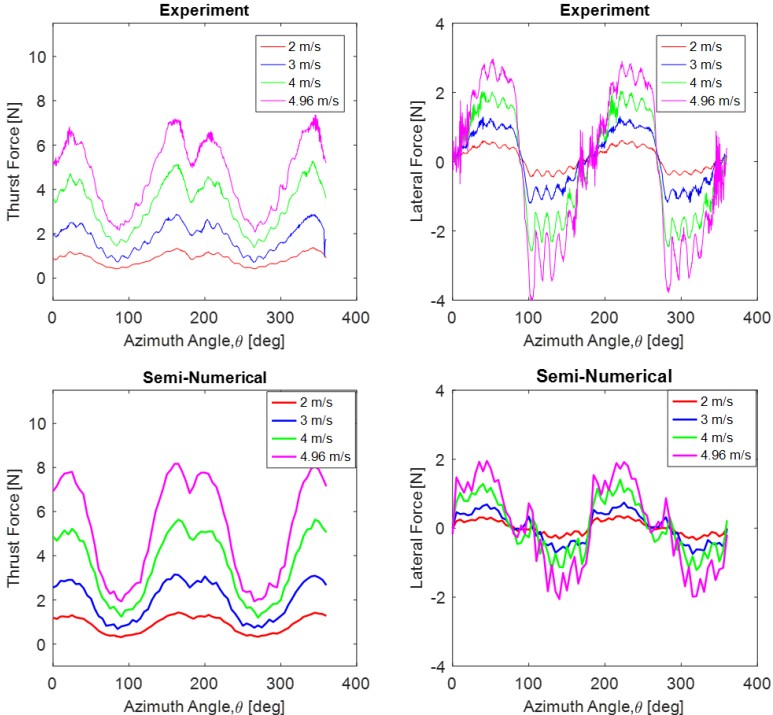

**Figure 17.** Validation of UTD semi-numerical parked load for two bladed turbine in the floating with wind only condition

This study experimentally investigates parked loads on floating VAWTs in a wind-wave basin. The study aims to provide
insights on the factors influencing parked loads. The study also aims to gather data to improve and validate a semi-numerical
parked load estimation tool for floating VAWTs under both wind-only and wind-and-wave conditions that is based on a vortex
aerodynamics model of the rotor and analytical model of tower drag. Further, this model is shown to capture the effects of
tower shadow, azimuth dependence and inertial effects..

The study presents data on parked loads across varying wind speeds, solidity, and operating conditions, examining the
impact of inertial loads from tilting and the correlation between tilt angles and parked loads. Validation of the semi-numerical
estimation tool is also included.

The findings highlight the significance of wind speed, solidity, azimuth dependence, and operating conditions in determining
parked loads. It was observed that VAWTs are subjected to substantial forces even in a stationary state, which necessitates robust
structural designs to ensure their durability and safety.

The insights gained from this study underscore the importance of considering parked loads in the design phase of VAWTs.
By incorporating the experimental data into the design process, engineers can develop more resilient turbines that can withstand
the stresses encountered during non-operational periods. This is particularly crucial for enhancing the longevity and reliability
of VAWTs, thereby making them a more viable option for renewable energy generation in diverse settings.The summary of the
paper can be stated as follows:



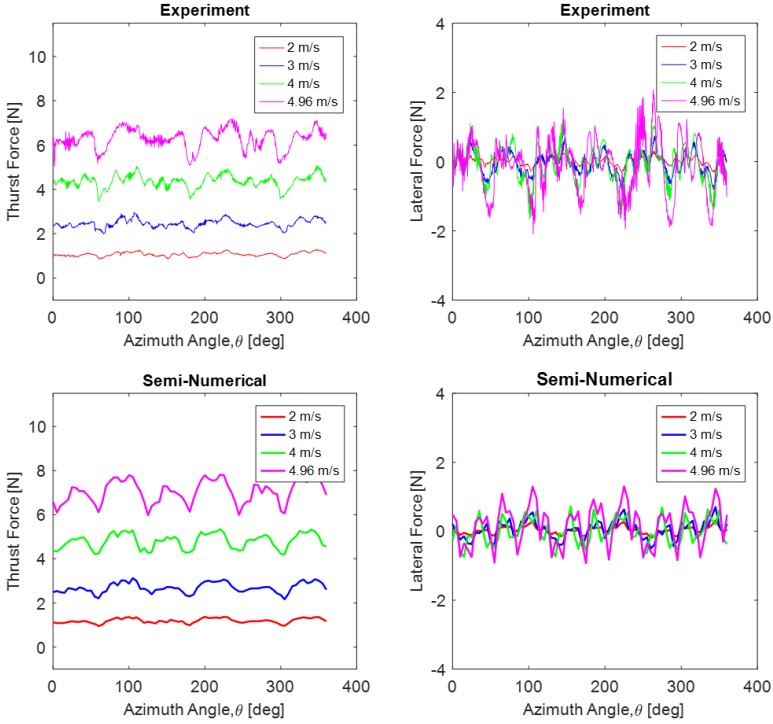

**Figure 18.** Validation of UTD semi-numerical parked load for three bladed turbine in the floating with wind only condition

– Solidity (in terms of number of blades) influences the parked loads. The load variation (range) decreases as the number of blades increases.

     – Average thrust load increases in floating condition due to inertial effect of turbine. Amplitude of lateral load also increases.

     – The floating wind-wave case exhibits more noisy loads due to dynamic nature compared to floating wind only case.

– The UTD's semi-numerical parked load tool well estimates the loads. Load magnitude, azimuth dependence, wind speed, solidity, and number of blades effects are well captured. Tower shadow is also captured.

     In conclusion, this study has advanced our understanding of the experimental parked loads on VAWTs and their impact on turbine performance. The results provide valuable guidelines for designing and implementing floating VAWTs that are both efficient and resilient. Future research should focus on further refining these findings through long-term field studies and

exploring innovative materials and design strategies to mitigate parked loads. By addressing these challenges, we can enhance the overall performance and adoption of VAWTs, contributing to the growth of sustainable wind energy solutions.





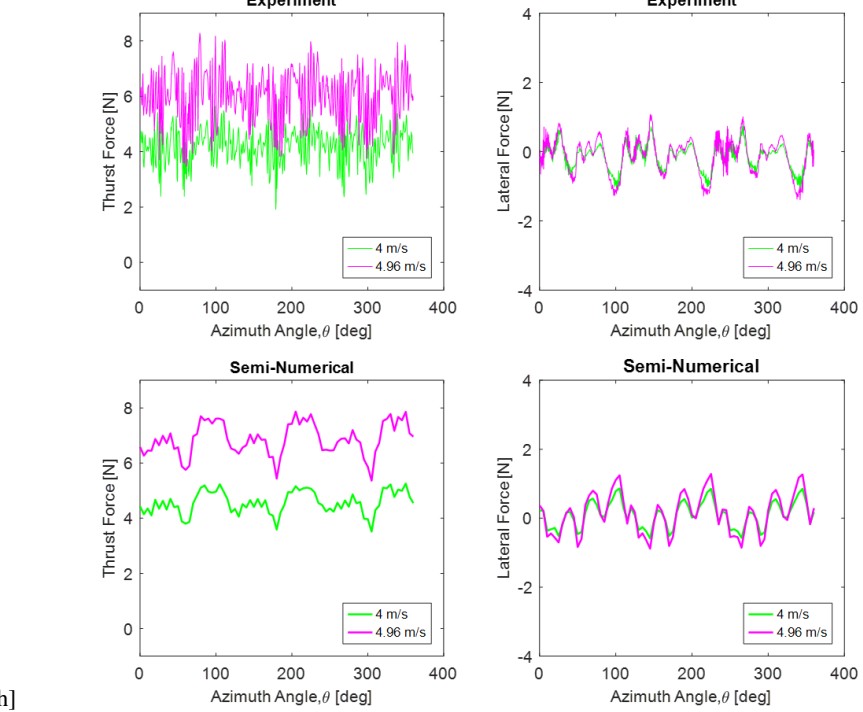

[h]

**Figure 19.** Validation of UTD semi-numerical parked load for 3B turbine in the floating tower base with wind and wave condition

*Author contributions.* This work is performed during the PhD of MSH under the supervision of DTG as part of an Advanced Re- search Projects Agency–Energy (ARPA-E)-funded project named A Low-cost Floating Offshore Vertical Axis Wind System. MSH and DTG contributed to the analysis and interpretation of the data, and the manuscript was prepared by MSH with the help of DTG.

*Competing interests.* The contact author has declared that nei- ther they nor their co-author has any competing interests.

*Acknowledgements.* The research presented herein was funded by the US Department of Energy Advanced Research Projects Agency-Energy (ARPA-E) under the ATLANTIS program with the project title "A Low-cost Floating Offshore Vertical Axis Wind System" associated with award no. DE-AR0001179. Any opinions, findings, and conclusions or recommendations expressed in this material are those of the authors and do not necessarily reflect the views of ARPA-E. The authors are grateful for the support of the ARPA-E program and staff, as
well as the project team.



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
