# Peer review of "Experimental Validation of Parked Loads for a Floating Vertical Axis Wind Turbine: Wind-Wave Basin Tests"

_Wind Energy Science, 2024_

## Author Comment (AC1)

The authors wish to thank the reviewer for the insightful and helpful comments. Below and in the revised manuscript (as indicated) the authors have responded point-by-point to each comment.

**Referee 2**

The manuscript presents results on the thrust and lateral forces of a laboratory-scale model of a Vertical Axis Wind Turbine (VAWT) tested in a wind-wave basin under parked conditions. Experimental data were compared with a semi-numerical code, which retrieves gravitational loads from the experimental measurements, in several cases: i) fixed-bottom – wind; ii) floating – wind – no waves; iii) floating – wind – waves. The goal of the work was to assess parked loads in relation to design parameters, such as the number of blades and free-stream velocity.

**Scientific Significance:**

The manuscript presents an original investigation using a laboratory-scale model to measure parked loads. Assessing forces in parked conditions is crucial for estimating the lifespan of a wind turbine in wind scenarios involving velocities higher than the cut-out wind speed. The authors also propose a semi-numerical code to predict the loads, which could be useful for designing full-scale turbines according to industry standards.

Response: Thank you for the insightful review and comments. The authors believe that addressing the comments below will enhance the quality of the paper.

**Scientific quality:**
Questions are addressed below in regards to scientific quality:
1. The authors propose a sensitivity analysis on the number of blades and wind speed. The effects of the number of blades on rotor loads have already been demonstrated in Sakib and Griffith (2022). The question that arises is: why should the number of blades have a greater impact on the design of a VAWT in parked loads than on the design of the turbine in operating conditions? Furthermore, why does the analysis focus solely on the number of blades, while the variation of solidity due to chord is disregarded? Please provide justification for these choices.
Response: Thanks for asking these valuable questions.

The authors have explored the effect of the number of blades on VAWT parked loads to study how the number of blades impact the azimuthal load variation in a revolution based on experiments, while Sakib and Griffith (2022) limited their study to numerical analysis only. A note has been added into the revised manuscript on lines 35–37.

Further, the load variation due to the solidity difference between the two-bladed and three-bladed turbines has also been examined, and the effect of solidity is shown to impact the magnitude of the loads (as expected). The difference in solidity arises from the design of the test bed to limit manufacturing complexity and cost. It was easy to

convert the two-bladed turbine into a three-bladed turbine using one set of blades. This note has been added to lines 93-94 of the revised manuscript.

2.  Is there any comparison in operating conditions to validate the numerical code against experimental data? The numerical model depends on data obtained from experiments. Is it not possible to formulate the model in such a way that it does not rely on experimental measurements? A model that does not depend on experiments would be useful for estimating loads without the need for additional testing. Furthermore, comparing numerical results with experimental data would serve as proper validation.
    Response: Thanks for raising these questions.

    Yes, it is possible to estimate the tilt angles (pitch and roll angles) in the pre-test and independent of the experiments. One can use the method developed in Gao et. al. [1] to predict the tilt angles. The input aerodynamic loads in this model can be supplied using CACTUS, while the hydrodynamic coefficients can be obtained from a commercial software like WAMIT [2] or an open source code like Capytaine[3]. However, predicting tilt data is out of scope of this study, but the approach and references noted provide a means of pre-test tilt motion prediction.

    We've added a note in lines 228-231 (along with relevant references) for pre-test modeling and prediction of the tilting response.

    In terms of the results presentation, we present the park loads comparison across different operating conditions (wind-wave-platform conditions) based on experimental data only. And later validated the semi-numerical park load data with experimental data across different operating conditions (wind-wave-platform conditions). Therefore, the comparison of semi-numerical loads across different operating conditions (test cases) seems unnecessary in addition to results already presented. Please note that the authors replaced the term "operating conditions" with "wind-wave-platform conditions" in the revised manuscript.

3.  Line 121: Please clarify the method used to obtain 'full-scale' wind speeds and the parameters the authors used to derive the range of 14.7 – 36.4 m/s according to Froude's law. It may be helpful to present the data in a table, showing both the scaled model and full-scale values.
    Response: Thanks for pointing this out. The method of full-scale wind speeds has been added to the updated Manuscript in lines 126-130.

    Authors used Froude scaling to convert the velocity with scale factor of $\lambda^{0.5}$ . Geometric comparison of the test turbine, which has 0.51 m radius, to the UTD 5 MW VAWT Sakib and Griffith (2022), which has 54.011 m radius, reveals that the test turbine is 1:104.87 scaled version of the full scale UTD 5 MW turbine. Applying Froude scaling with a velocity scale factor of $\lambda^{0.5}$ results in a wind speed range of 20.48 ms-1 to 50.79 ms-1 for full-scale 5 MW UTD VAWT.

In the previous manuscript, the scale factor of the test turbine was incorrectly calculated as 54 instead of 104.87, leading to an incorrect wind speed range of 14.7–36.4 m/s. This has been corrected in the updated manuscript. However, it is important to note that the tested model is not a scaled version of UTD's 5 MW turbine. The wind speed scaling is presented solely to compare the wind speed selections in the test campaign to those at the 5MW scale.

4. Line 137: Are the values for wave period and height representative of full-scale scenarios?
Response: Yes, they are representative of full-scale scenarios of IEA 15 MW wind turbines. The full scale height and period are 10.85 m and 13.5 s, respectively [4]. This information has been added to the updated manuscript in lines 146-147.

5. Please clarify the sentences in line 143: "No wake effects have been included…" and line 147: "Blade wake […] were considered…". Two different wake effects are mentioned, one of which is included while the other is not. Please provide further explanation. Induction due to the wake should be accounted for anyway: also VAWT in steady conditions produces blockage caused by the wake despite the rotor is still.
Response: Thanks for raising this important point.
Induction due to blade wake was neglected as it is too small compared to freestream velocity and to make the model simpler. However, induction due to tower wake was included as it is easy to integrate. The updated manuscript noted these points in lines 165-167 and 170-171.

6. Dynamic stall is not considered, nor are unsteady effects in the case of wind and wave excitation. Have the reduced frequencies been evaluated to demonstrate that unsteady aerodynamics does not occur?
Response: Thanks for the comment. As the turbine is parked, the chance of occurring dynamic stall is very minimal. Therefore, we did not consider the dynamic stall model in the semi-numerical tool.

The authors have analyzed the reduced frequency in the updated manuscript. Reduced frequency is a parameter used to determine whether the inflow is unsteady, and it can be defined as follows [5]:

$$K_i = \frac{\omega_{ptfm}\, C_i}{2\sqrt{(U_\infty^2 + r_i^2 \Omega_\infty^2)}}$$

where, $\omega_{ptfm}$ is the platform frequency, $C_i$ is airfoil chord, $U_\infty$ is freestream velocity, $r_i$ is section radius, $\Omega$ is rotational velocity of rotor.

[Figure]

*Figure 1: Reduced frequency for different platform and turbine operating conditions. Note: Section Radius refers to the rotor radius, which varies with rotor height.*

Reduced frequency does not go above a value of 0.05 for both floating wind only and floating wind wave condition for any of the wind speeds, as shown in Figure 1. The highest reduced frequency (K) of 0.0158 occurs in the 3B floating wind wave pitch case. As wind speed increases, the reduced frequency decreases. Specifically, for the 3B floating wind wave pitch case, the reduced frequencies (K) at wind speeds of 2, 3, 4, and 4.96 m/s are 0.0158, 0.0105, 0.0079, and 0.0063, respectively. This indicates that higher wind speeds

result in a steadier inflow. Therefore, the inflow can be considered as steady. Notes have been added on lines 154-165 to provide this information.

7. Equations 1 and 2 are identical.
Response: Thanks for pointing this out. Equation 2 is related to tangential velocity ($V_T$). The equation has been modified and added in line 173.
Modified equation:
$V_T = T_X U_X + T_Y U_Y + T_Z U_Z$

8. How have the polars been adjusted to extend their applicability to VAWT operations, where angles of attack can reach very high values (especially in parked conditions, where peripheral speed is nearly zero)?
Response: Thanks for pointing this out. The airfoil polars are applicable to VAWT operations. The airfoil polar is supplied with CACTUS code. The angle of attack ranges from -180 to +180 degrees. The details of how polars were calculated can be found in [6]. The note is also presented in manuscript in lines 152-153.

9. Inertial effects are defined as the projection of gravitational loads on the structure. Shouldn't proper inertial effects be considered when the turbine is subjected to both wave and wind forces? The entire structure experiences motion, with associated velocities and accelerations that affect the aerodynamics. Consider using a different term instead of "inertial effects," as this might be interpreted as including inertia forces in the model.
Response: Thanks for pointing out another important issue. Authors replaced the term "inertial effects" with "gravitational effects" to make the point clear.

10. Figure 8: The use of 'uncertainty' bars is unclear. According to the authors, the variability of thrust and lateral forces is attributed to azimuthal positions. Are the uncertainty bars representing the amplitudes over a period?
Response: Thanks for this comment. The bars do not represent uncertainty; rather, they represent the azimuthal variation of loads. A short description has been added in the manuscript to make the point more clear [line 264-265].

11. A general question regarding experimental data: Have the authors assessed the uncertainty in the measurements?

Response: We have addressed the uncertainty in the measurements; for example, as shown in Figure 2 below and in Figure 8 on page 14 in the revised manuscript. Subsection 3.1.1 has been added to the revised manuscript for presentation of the uncertainty analysis [7]:

$$U = \sqrt{\beta^2 + (t\sigma)^2}$$

where, *U* is resultant uncertainty and *t* is the coverage factor of T distribution. The value of *t* is 1.96 at 95% confidence level.

[Figure]

*Figure 2: Parked loads with uncertainty of 2B turbine for locked wind only condition.*

12. Figures 10, 11, and 12: The comparisons at different wind speeds in separate plots make it harder to understand the results. Consider merging them into a single plot or showing the effect of wind speed in one plot and the effect of the floating motion in another (e.g., in Figures 10 and 11). Additionally, consider discretizing the x-axis in increments of 90 degrees.

Response: Thanks for the suggestions. The author carefully reviewed your suggestions. However, we believe that combining all wind speeds into a single plot would make the figure overly crowded and harder to interpret. The primary goal of Figures 10 and 11 (Figure 11 and 12 in the updated manuscript) is to compare results across different operating conditions (test cases), such as locked wind only, floating wind only, and floating wind with wave cases. The current plotting approach effectively serves this purpose.

13. Figures 14, 15, 17, and 19: If the purpose is to compare measurements with numerical data, these comparisons should be shown in the same plot for clarity, for each wind speed.

Response: Thanks for the suggestions. The authors have merged the experimental and semi-numerical data into the same plot for easier comparison. The change can be seen in Figures 15, 16,18, 19, and 20. New figures are also added to provide the non-dimensional results as well.

14. Can the authors clarify how the data over one rotation have been considered (e.g., in Figure 19)? Is the force over the θ angle an average of several rotations or the last period? It could be useful to present the phase-averaged data.

Response: The experimental data represents the phase average of 5 revolutions. The semi-numerical model data is based on the last revolution values after the CACTUS simulation has converged. These points are noted in line 331-332.

**Presentation quality**

Please find a list of typos or request of clarification in the manuscript:

1. Line 39: the authors refer to HVAWTs, it is suggested to call H-shape/H-shaped since HVAWT might be confusing (between HAWT and VAWT).
   Response: Thanks for pointing out that. Authors understand that "HVAWT" and "HAWT" might seem similar. Now we have changed "HVAWT" to "H-VAWT". And the sentence already mentioned straight-bladed vertical axis wind turbine is termed as HVAWT. Which should clarify the confusion between "H-VAWT" and "HAWT".

2. Line 61: a instead of an
   Response: Authors replaced "an" with "a" in line 63.

3. Line 79: in instead of on
   Response: Authors did put " in" instead of "on" in the line 83 too.

4. Line 98 and Line 117: refer to the table 2 in the previous sentence, so it is clear that is connected to the concept explained.
   Response: The referring sentences are moved to the previous sentence on line 101 and line 122 to make it connected to the concept.

5. Line 128: is the operating condition
   Response: Authors inserted "the" in between.

6. Line 194: typo in " for are".
   Response: Thanks. We fixed the typo. Updated sentence: "The measured park loads are in the updated sensor measurement load direction."

7. Line 200: weight
   Response: thanks, replaced 'wight' with "weight".

8. Figure 9. The figure has no legend. Please provide for better understanding
   Response: Thanks for raising this issue. However, authors think that legend in this figure would be redundant.

9. Line 236-239: repetition
   Response: Removed the repeated sentence.

10. Line 263: a dot is missing
    Response: Thanks for mentioning this. Authors fixed the issue.

11. Line 317: delete a '.'
    Response: Thanks! We deleted the extra '.'.

**References**

1. Gao, J., Griffith, D.T., Sakib, M.S., and Boo, S.Y. (2022). *A semi-coupled aero-servo-hydro numerical model for floating vertical axis wind turbines operating on TLPs*. Renewable Energy *181*, 692–713. https://doi.org/10.1016/j.renene.2021.09.076.

2. Lee, C. H. and Newman, J.N. (2006). *WAMIT® User Manual, Versions 6.3, 6.3PC, 6.3S, 6.3S-PC* at WAMIT, Inc.

3. Ancellin, M., and Dias, F. (2019). *Capytaine: a Python-based linear potential flow solver*. J. Open Source Softw. *4*, 1341. https://doi.org/10.21105/joss.01341.

4. Fowler, M.J. (2023). *Floating Offshore-wind and Controls Advanced Laboratory Floating Offshore-wind and Controls Advanced Laboratory Program: 1:70-scale Testing of a 15 Mw Floating Wind Turbine Program: 1:70-scale Testing of a 15 Mw Floating Wind Turbine*.

5. Matha, D., Cruz, J., Masciola, M., Bachynski, E.E., Atcheson, M., Goupee, A.J., Gueydon, S.M.H., and Robertson, A.N. (2016). *Modelling of Floating Offshore Wind Technologies* https://doi.org/10.1007/978-3-319-29398-1_4.

6. Sheldahl, R., and Klimas, P. (1981). *Aerodynamic characteristics of seven symmetrical airfoil sections through 180-degree angle of attack for use in aerodynamic analysis of vertical axis wind turbines*.

7. Coquilla, R. V, Obermeier, J., White, B.R., Coquilla, R. V, Obermeier, J., and White, B.R. (2007). *Calibration Procedures and Uncertainty in Wind Power Anemometers*. *44*.

---

## Author Comment (AC3)

The authors wish to thank the reviewer for the insightful and helpful comments. Below and in the revised manuscript (as indicated) the authors have responded point-by-point to each comment.

**Referee 1**

The study evaluates the aerodynamic parked loads of a model-scale floating troposkein VAWT in a wind-wave basin under different conditions: a fixed tower base, floating without waves, and floating with waves. The influence of wind speed, solidity (via blade count variation), and rotor azimuth on the parked load is analyzed. In general, the paper presents a good approach and addresses an important aspect of floating VAWTs, an area that remains underexplored in the literature. However, several comments are provided below to enhance the quality of the paper, particularly in terms of the findings and discussions:

Response: Thank you for the insightful review and comments. The authors believe that addressing these suggestions has enhanced the quality of the paper.

**Methods:**
1. The authors highlight that using static airfoil polars is reasonable in the context of parked loads. However, this assumption holds only for the fixed-base system. For the other two cases (with the floating system and with floating system plus waves), the variation in the angle of attack could occur at a frequency high enough to impose unsteady load conditions. This aspect should be further discussed.

Response: Thanks for pointing out this critical issue. Use of static airfoil polars makes sense for the fixed-base system. For the two floating cases, we've performed some analysis of reduced frequency to address the point noted by the reviewer.
Reduced frequency is a parameter used to determine whether the inflow is unsteady and it can be defined as follows [1]:

$$K_i = \frac{\omega_{ptfm}\, C_i}{2\sqrt{(U_\infty^2 + r_i^2\, \Omega^2)}}$$

where, $\omega_{ptfm}$ is the platform pitching frequency, $C_i$ is airfoil chord, $U_\infty$ is freestream velocity, $r_i$ is section radius, $\Omega$ is rotational velocity of rotor.

According to Theodorsen's theory, a flow can be categorized as unsteady if $K > 0.05$. The reduced frequency ($K$) does not go above a value of 0.05 for both floating wind only and floating wind wave conditions for any of the wind speeds, as shown in Figure 1. The highest reduced frequency (K) of 0.0158 occurs in the 3B floating wind wave pitch case. As wind speed increases, the reduced frequency decreases. Specifically, for the 3B floating wind wave pitch case, the reduced frequencies (K) at wind speeds of 2, 3, 4, and 4.96 m/s are 0.0158, 0.0105, 0.0079, and 0.0063, respectively. This indicates that higher wind speeds result in a steadier inflow. Therefore, the inflow can be considered as steady. Notes have been added on lines 154-165 to provide this information.

[Figure]

*Figure 1: Reduced frequency for different platform and turbine operating conditions. Note: Section Radius refers to the rotor radius, which varies with rotor height.*

2. There is a typo in equations (1) and (2); both equations are identical. Please correct this.

Response: Thanks for pointing this out. Equation 2 is related to tangential velocity ($V_T$). The equation has been modified and added in line 173.
Modified equation:
$V_T = T_X\,U_X + T_Y\,U_Y + T_Z\,U_Z$

3. Why not nondimensionalize all the parked load forces or even normalize them with respect to the rated forces? This would make it easier for readers to compare and interpret the results.

Response: Thanks for raising the point about normalization. Authors included both the dimensional and nondimensional parked loads in this version.

The nondimensionalized parked forces are defined as follows.

$$C_{Th} = \frac{F_{Th}}{\frac{1}{2}\rho\,A\,U_\infty^2}$$

$$C_{Lat} = \frac{F_{Lat}}{\frac{1}{2}\rho\,A\,U_\infty^2}$$

Where, $C_{Th}$ is thrust force coefficient, $C_{Lat}$ is lateral force coefficient, $F_{Th}$ is thrust force, $F_{Lat}$ is lateral force, $\rho$ is density of air, $A$ is the rotor area, and $U_\infty$ is freestream velocity.

The revised manuscript shows both nondimensional and dimensional results. The nondimensional results are shown in Figures 15a, 17a, 18a, 19a, and 20 a for 2B locked with wind only, 3B locked with wind only, 2B floating with wind only, 3B floating with wind only, and 3B floating with wind and wave conditions, respectively.

[Figure]

*Figure 2: Nondimensional parked loads of 3B turbine for locked wind only condition.*

[Figure]

*Figure 3: Nondimensional parked loads of 2B turbine for locked wind only condition.*

[Figure]

*Figure 4 : Nondimensional parked loads for 2B turbine for floating wind only condition.*

[Figure]

*Figure 5:  Nondimensional parked loads for 3B turbine for floating wind only condition.*

[Figure]

*Figure 6: Nondimensional parked loads of 3B turbine for floating wind wave condition.*

**Results and Discussion:**

4. Why not compare the experimental measurements directly against the UTD semi-numerical model, instead of presenting them in two separate graphs? How can the reader assess the accuracy of the numerical model if the results are not directly compared?

Response: Thanks for mentioning direct comparison between experimental and semi-numerical results. Authors reproduced the plots in the same graphs as follows and are now presented for non-dimensional and dimensional cases.

The direct comparison between experiment and semi-numerical model results for dimensional results are presented in the revised manuscript in Figures 15b,17b, 18b, 19b, and 20b for 2B locked with wind only, 3B locked with wind only, 2B floating with wind only, 3B floating with wind only, and 3B floating with wind and wave conditions, respectively.

[Figure]

*Figure 7: Comparison between experimental and semi-numerical parked loads for locked wind only tower base conditions of 2B turbine. The continuous lines represent experimental loads, where the dashed lines represent semi-numerical loads.*

[Figure]

*Figure 8: Comparison between experimental and semi-numerical parked loads for locked wind only tower base conditions of 3B turbine. The continuous lines represent experimental loads, where the dashed lines represent semi-numerical loads.*

[Figure]

*Figure 9: Comparison between experimental and semi-numerical parked loads for floating wind only tower base conditions of 2B turbine. The continuous lines represent experimental loads, where the dashed lines represent semi-numerical loads.*

[Figure]

*Figure 10: Comparison between experimental and semi-numerical parked loads for floating wind only tower base conditions of 3B turbine. The continuous lines represent experimental loads, where the dashed lines represent semi-numerical loads.*

[Figure]

*Figure 11: Comparison between experimental and semi-numerical parked loads for floating wind wave tower base conditions of 3B turbine. The continuous lines represent experimental loads, where the dashed lines represent semi-numerical loads.*

5. I believe it is crucial to show the influence of tower tilting on the angle of attack variation, either in the case of floating alone or floating with waves. These changes are likely to impact the estimation of the aerodynamic loads and should be addressed in the analysis (at least reporting the values might give a reader a sense on how unsteady are the loads).

[Figure]

*Figure 12: Variation of angle of attack with respect to azimuth for 1st blade of 2B turbine. The compared cases are no tilt, 2 deg pitch, and 2 deg roll.*

Response: In the experiment we did not record any data for angle of attack. However, authors would like to show tilting effect on angle of attack for the case of a static pitch angle of 2 deg (average pitch at 4.96 ms$^{-1}$ for floating wind only case). The influence of tilting on angle of attack is very minimal which is shown in **Error! Reference source not found.**. The data presented here corresponds to the 1$^{st}$ blade of 2B turbine at 4.96 ms$^{-1}$, where the 0 ° azimuthal position represents the blade's direct wind ward position.

6. The general division of sections could be improved. In both the abstract and the introduction, the study compares the fixed tower system with the floating system and the floating system with waves, highlighting how results vary as complexity increases. However, throughout the results section, these comparisons are not directly made. It would be beneficial to address these differences directly within the results section.

Response: Thanks for pointing out the issue of general division of section. Section 3.1.4 of the initial manuscript provides description and comparison of the parked loads among different operating conditions (wind-wave-platform conditions) such as locked with wind only, floating with wind only, and floating with wind wave conditions.

7. In case of floating tower with wind and wave: "the numerical model neglects the dynamic nature due to floating wind and wave effect" please elaborate with more details, why the model neglects these effects, what are the implications on the model predictions and how can it improved?

Response: The existing model is not formulated to predict the dynamic nature of pitch and roll motions and respective dynamic thrust and lateral loads due to coupled wind, wave, and floating platform effects, although it does handle static pitch and roll motions. A prediction of pitch and roll motions requires integrating aerodynamic, wave, hydrodynamic, platform and mooring dynamic models. A code such as WAMIT [2], or an open source code Capytaine [3], can be used to model the hydrodynamics and then coupled with mooring and aerodynamic models to predict the platform dynamics [4]. Such a model can be used for pre-test motion prediction. Here, the authors have restricted their study to the current semi-numerical parked load model and comparison with the experimental data. Ongoing research is focused on developing a coupled wind-wave-floating dynamics model for both parked and operating cases, which will be presented in the future. This note has been added to the manuscript in lines 378-384.

**Conclusions:**
8. The statement in the conclusion that the numerical model is validated and optimized is unclear, as the numerical model was not directly compared with experimental measurements throughout the paper. Additionally, other data from the numerical model, such as the variation in the blade angle of attack, should be presented. This is important for assessing the assumption that unsteady load corrections were not needed for the aerofoil polars.

Response: Thanks for those insightful comments. The data from the numerical model is directly compared with the experimental data in the revised version of the paper. The comparison shows that the semi-numerical tool well predicts the magnitude of parked loads, azimuthal dependence on loads, and the effects of wind speed, and solidity for all the operating conditions. Additionally, this tool accurately captures the tower shadow at the 180° azimuthal location. However, the model is not formulated to predict the dynamic nature of pitch and roll motions and respective dynamic thrust and lateral loads due to coupled wind, wave, and floating platform effects.

Due to limitation of the CACTUS tool, we were unable to analyze the variation of angle of attack (AOA) for dynamic tilting cases of the turbine. However, we conducted a comparison of the angle of attack by statically tilting the turbine at 2 degrees—representing the mean pitch angle at 4.96 m/s for the floating wind-only condition—against the no-tilt case. While the middle element showed no variation, the top element exhibited slight changes. This behavior occurs because, for a parked turbine, the AOA is solely influenced by the free-stream velocity, with induced velocity and rotational velocity assumed to be zero.

We also performed a reduced frequency analysis for the floating platform cases. The analysis shows that the inflow is not unsteady. Therefore, the assumption of 2D static airfoil makes sense.

9. The paragraphs starting at lines 359 and 364 are identical and appear to be replicated by mistake.

Response: Thanks for mentioning this. Authors removed the replicated paragraph.

10. In the wave+floating parked load scenario, the numerical model has some drawbacks as reported in the results section. This issue is not highlighted in the conclusions.

Response: Thanks for pointing out this. Authors really appreciate this point. We added the drawbacks also in the Conclusions in the 4th bullet point of the summary of the paper lines 414-416.

11. The sentence "This study has advanced our understanding of the experimental parked loads on VAWTs and their impact on turbine performance" seems to describe the main objective of the paper. The numerical model, while useful, should be considered a tool to gain insight into unmeasurable quantities after validating the load measurements. It might be better to frame this as the central goal of the study.

Response: Thanks for outlining the main theme and offering a thoughtful suggestion. The authors believe that the paper is already built on "advancing our understanding of experimental parked load of VAWTs" theme. To further clarify that the abstract and introduction sections are slightly revised. Additionally, the statement" semi-numerical park load model is a tool to gain insight into unmeasurable quantities"  has been added to conclusion of abstract (line 16) and to the main contribution part of introduction (line 65).

**References**

1. Matha, D., Cruz, J., Masciola, M., Bachynski, E.E., Atcheson, M., Goupee, A.J., Gueydon, S.M.H., and Robertson, A.N. (2016). *Modelling of Floating Offshore Wind Technologies* https://doi.org/10.1007/978-3-319-29398-1_4.
2. Lee, C. H. and Newman, J.N. (2006). *WAMIT® User Manual, Versions 6.3, 6.3PC, 6.3S, 6.3S-PC* at WAMIT, Inc.
3. Ancellin, M., and Dias, F. (2019). *Capytaine: a Python-based linear potential flow solver*. J. Open Source Softw. *4*, 1341. https://doi.org/10.21105/joss.01341.
4. Gao, J., Griffith, D.T., Sakib, M.S., and Boo, S.Y. (2022). *A semi-coupled aero-servo-hydro numerical model for floating vertical axis wind turbines operating on TLPs*. Renewable Energy *181*, 692–713. https://doi.org/10.1016/j.renene.2021.09.076.

---

## Author Comment (AC4)

[revised manuscript text omitted]

To date, very few parked load studies of VAWTs are found in the literature. Sakib and Griffith (2022) analyzed parked aerodynamic load for a 5 MW conceptual VAWT, considering rotor design variables such as tapered blade chord, number of blades, aspect ratio (the ratio of rotor height to diameter). Although Sakib and Griffith (2022) numerically studied the effects of the number of blades on parked loads, in this study we have studied the effects of the number of blades on azimuthal variation of parkloads both numerically and experimentally. The analytical tool used in Sakib et al's study is validated here and improved by adding the capacity for estimating parked loads for offshore floating turbines. Ottermo et al. (2012) developed an analytical model to estimate extreme loads under parked conditions. Paulsen et al. (2013) conducted a CFD study to predict both the operating and parked load for the Deep Wind concept. Kuang et al. (2019) performed a numerical CFD investigation on the flow characteristics and dynamic responses of a parked straight-bladed vertical axis wind turbine (H-VAWT) and concluded that pressure distribution on the upwind blade surface are similar at different azimuthal locations. As wind speed increases, turbulent flow characteristics and wake effects become more pronounced, while dynamic responses due to parked conditions can be neglected. The only experimental parked load analysis of 12 kW VAWT was done by Goude and Rossander (2017), which used fixed base (locked platform) for an H-VAWT.

This study focuses on the experimental investigation of parked loads on vertical axis wind turbines with experiments performed in a wind-wave basin. The research aims to provide a comprehensive understanding of the factors affecting parked

loads and their impact on turbine performance. The findings from this study offer valuable insights for the design, operation, and maintenance of VAWTs, ultimately contributing to the advancement of sustainable wind energy technologies.

50     This paper also focuses on enhancing the capacity of UTD's (UT-Dallas) existing semi-numerical VAWT parked load estimating tool originally developed by Sakib and Griffith (2022). This tool make use of a mid-fidelity, open-source, free vortex method code CACTUS (Code for Axial and Cross-flow Turbine Simulation) [Murray and Barone (2011)] and analytical methods. CACTUS was developed by Sandia National Laboratories in FORTRAN 95 language using VDART3 [Strickland et al. (1980)] code as a basis. CACTUS code is capable of performing an analysis of any arbitrary turbine configuration by seg-

55 menting the turbine blades and struts into a set of blade elements. More detailed explanation of CACTUS can be found in Lu (2020).

    In summary, the aim of this work is to experimentally study and validate the modeling of parked loads for Darrieus VAWTs with different platform conditions and to enhance the capacity of UTD's existing VAWT parked load estimating tool. The main contributions of this work can be summarized as below:

60     – An experimental study is performed that examines troposkein wind turbines with two blades (2B) and three blades (3B).

    – The parked dataset presented here is unique because the data includes both fixed base (locked platform, zero tilting), and for floating platform conditions (floating with and without waves).

    – Experimental data is gathered to validate a semi-numerical parked load estimating tool, originally developed by Sakib and Griffith (2022). The tool has also been improved to assess the parked load for offshore floating turbines with tilting.

65     Ultimately this tool can be used to gain insights into the unmeasurable quantities.

    Section 2 covers the methods for experiments, the test campaign, and the development of the parked loads model. Section 3 presents results and discussions for the experimental measurements and model validation efforts, with a focus on the effects of wind speed, turbine solidity (varying number of blades), different platform conditions, and effect of rotor azimuth. Section 4 presents the concluding remarks.

**2   Methods**

70   The VAWT rotors were tested at UMaine's Alfond Wind-Wave Ocean Engineering Laboratory (W2) [Cole et al. (2017)]. This unique facility is equipped with high-precision measurement instruments and allows for variable water depths, wind, and wave conditions. The basin measures 30 meters in length and 9 meters in width, with a maximum water depth of 5 meters. The wind machine, whose dimension is 7 m x 3.5 m, can generate up to 12 $ms^{-1}$ wind speed using a narrower condenser passage;

75 however, during these VAWT tests, maximum wind speeds were limited to 4.96 $ms^{-1}$ with a turbulence intensity of 3.9%, since the condenser was removed. A detailed description of the testing facility can be found in Parker (2022) .

[Figure]

**Figure 2.** Visual of the 3B test turbine configuration CAD model with dimensions

**2.1 Test turbines**

[revised manuscript text omitted]

(a) Fixed base      (b) Floating base

**Figure 4.** Tower base options used in the experiments.

125     The second variable we considered is wind speed. The test facility is capable of producing a maximum of 4.96 ms$^{-1}$ wind speed. Therefore, the turbines were tested at wind speeds varied between 2 ms$^{-1}$ to 4.96 ms$^{-1}$ keeping the turbine rotational speed constant at 1 RPM. Geometric comparison of the test turbine (having radius of 0.51 m) to the UTD 5 MW VAWT [Sakib and Griffith (2022)] (having radius of 54.011 m) reveals that the test turbine is a 1:104.87 scaled version of the full-scale UTD 5 MW turbine. However, the test turbine is not a scaled design of the UTD 5 MW VAWT; this comparison is made solely to 130 estimate the full-scale wind speeds. Applying Froude scaling with a velocity scale factor of $\lambda^{0.5}$ results in a wind speed range of 20.48 ms$^{-1}$ to 50.79 ms$^{-1}$ for full-scale 5 MW UTD VAWT. This wind speed range is reasonable because the site specific 50 years return period having a 10 minutes average wind speed of 30.96 ms$^{-1}$ [Sirnivas et al. (2014)] falls in between this range.

    The third variable considered in this test campaign is the number of blades. The two-bladed turbine (2B) has solidity of 0.194, whereas the three-bladed turbine (3B) has solidity of 0.291. The geometric configurations and visual representation of

**Table 3.** Test matrix

| Variables | Short Description |
| --- | --- |
| Azimuth | Azimthal position ranges from 0° to 360°. The position of first blade in the wind ward direction represents the 0° azimuthal position. |
| Wind Speed | Wind speeds are varied between 2 and 4.96 ms⁻¹. |
| Number of Blades | Two-bladed (2B) and three-bladed (3B) troposkein VAWTs are tested in this campaign. 2B turbine and 3B turbine have solidities of 0.194 and 0.291, respectively. |
| Twind-wave-platform conditions | Three wind-wave-platform conditions were considered in the test campaign. Those are locked with wind only, floating with wind only, and floating with wind and wave cases. |

tested turbines are shown in Table 1 and Figure 2, respectively. The detailed design and manufacturing of the turbines can be found in Hossain et al. (2024).

The fourth variable we considered is the wind-wave-platform conditions. The turbines were tested for three wind-wave-platform conditions: a) locked (fixed tower base) with wind only, b) floating with wind only (no waves), and c) floating with wind and waves. The locked with wind only condition represents a case where the tower base is attached to a stinger to restrict tilting motion (i.e., fixed base), and the turbine is rotated only in presence of wind with no waves present. Please see Figure 4 for details of the tower base attached to the stinger. The floating with wind only condition makes use of a semi-submersible floating platform. The column of floating platform is connected to horizontal mooring lines (not shown in the Figure 4) to keep the floating platform in place. However, the mooring lines were not connected to the basin bed. Instead, they were connected to the side wall to reduce the movement for this specific wind wave basin. In this condition, the turbines are only exposed to the wind. The floating wind and wave condition also uses the semi-submersible floating platform; however, the turbines are exposed to both wind and waves. In this test campaign, a regular wave with a height of 0.155 m and a period of 1.61 s is used, which corresponds to a full-scale IEA 15 MW horizontal axis wind turbine with a height of 10.85 m and a period of 13.5 s [Fowler (2023)]. Readers are referred to Figure 4 for the visuals of locked and floating platforms and referred to Table 3 for the test matrix used in this campaign.

**2.4 UTD's semi-numerical parked load tool**

A semi-numerical parked load tool is developed to estimate the parked loads of the tested turbines. As methods like CFD will be very computationally intensive to predict parked loads, a semi-numerical method has been developed with goals of accuracy and low computational effort. This tool makes use of static airfoil polar supplied with the CACTUS tool. Airfoil polars are applicable to VAWT operations, and they range from -180° to 180° of angle of attack. This tool makes use of static

airfoil polar supplied with the CACTUS tool. Use of static airfoil polars makes sense for the fixed-base system. For the two floating cases, we've performed analysis of reduced frequency to see if the inflow is steady or unsteady. Reduced frequency is a non-dimensional number, and it is defined in Matha et al. (2016) as follows:

$$K_i = \frac{\omega_{\text{ptfm}} C_i}{2\sqrt{U_\infty^2 + r_i^2 \Omega^2}} \tag{1}$$

where, $\omega_{ptfm}$ is the platform pitching frequency, $C_i$ is airfoil section chord, $U_\infty$ is freestream velocity, $r_i$ is section radius, $\Omega$ is rotational velocity of rotor. According to Theodorsen theory, a flow can be categorized as unsteady if reduced frequency $K$ in Equation 1 exceeds 0.05. The reduced frequency ($K$) does not go above a value of 0.05 for both floating wind only and floating wind wave conditions for any of the wind speeds for either 2B or 3B wind turbines. The highest reduced frequency ($K$) of 0.0158 occurs in the 3B floating wind wave pitch case. As wind speed increases, the reduced frequency decreases. Specifically, for the 3B floating wind wave pitch case, the reduced frequencies ($K$) at wind speeds of 2, 3, 4, and 4.96 m/s are 0.0158, 0.0105, 0.0079, and 0.0063, respectively. This indicates that higher wind speeds result in a steadier inflow. Therefore, the inflow can be considered quasi-steady, and the use of static polar for floating cases also makes sense. Since the induction due to blade wake is small compared to freestream velocity, and to make the model simpler, blade wake effects have been excluded from the model. Moreover, strut effects, and finite aspect ratio corrections have also not been considered.

The local relative velocity and angle of attack for all the azimuthal locations were calculated using CACTUS. In the relative velocity calculation, we only considered local free-stream velocity ignoring the rotational and induced velocity components because the rotor is standstill for the parked condition. However, induction due to tower wake was considered in the model as it was easy to implement in the model.

$$V_N = N_x U_x + N_Y U_Y + N_Z U_Z \tag{2}$$

$$V_T = T_x U_x + T_Y U_Y + T_Z U_Z \tag{3}$$

where the $V_N$ and $V_T$ are the normal and tangential velocity. $U$ is the freestream velocity. $N$, and $T$ are normal vector component and tangential vector component, respectively. The value of normal vector component ($N$) and the tangential vector component ($T$) are calculated using the 3D vortex-based code CACTUS. The directions of forces and velocities are shown in Figure 5. Where the 0° azimuth is in wind ward direction. The thrust load ($F_{Th}$) is in wind ward direction and the lateral load ($F_{Lat}$) is normal to the thrust load. The normal force ($F_N$) is toward the center and tangential force ($F_T$) is in the tangential direction of airfoil. Local relative velocity is calculated as,

$$V_R = \sqrt{[V_N]^2 + [V_T]^2} \tag{4}$$

After that, the angle of attack ($\alpha$) is calculated from the normal and tangential components of the velocity.

$$\alpha = [Tan]^{-1}(V_N/V_T) \tag{5}$$

Then the lift ($C_L$) and drag ($C_D$) coefficients for respective angle of attack are calculated using the static airfoil polar supplied with CACTUS tool. After that, the local normal ($C_N$) and tangential ($C_T$) force coefficients are calculated using $C_L$ and $C_D$

[Figure]

**Figure 5.** VAWT forces and velocities

as follows:

$$C_N = C_L \, Cos(\alpha) + C_D \, Sin(\alpha) \tag{6}$$

$$C_T = C_L \, Sin(\alpha) - C_D \, Cos(\alpha) \tag{7}$$

where $C_L$ represents lift coefficient, and $C_D$ represents drag coefficient. The normal and tangential force coefficients are local, in other words with respect to the specific element.

Therefore, normal and tangential force coefficients need to be re-referenced to full turbine scale to calculate the thrust and lateral forces. The conversion is done using the following equations.

$$C_{FtN} = C_N \, (A_E/A) \, (V_R/U)^2 \tag{8}$$

$$C_{FtT} = C_T \, (A_E/A) \, (V_R/U)^2 \tag{9}$$

where $A_E$ is element area, $A$ is rotor area, $C_{FtN}$ and $C_{FtT}$ are normal and tangential force coefficients w.r.t. rotor, respectively. After that thrust ($C_{Th}$) and lateral ($C_{Lat}$) force coefficients are calculated applying specific normal and tangential directions in the azimuthal locations.

$$C_{Th} = N_x \, C_{FtN} + T_x \, C_{FtT} \tag{10}$$

$$C_{Lat} = N_Y \, C_{FtN} + T_Y \, C_{FtT} \tag{11}$$

[Figure]

**Figure 6.** Schematic of gravitational effect on floating VAWT due to tilting.

Thrust and lateral loads for locked platform case,

$$F_{Th} = 1/2\,\rho A\,C_{Th}\,U^2 \tag{12}$$

$$F_{Lat} = 1/2\,\rho A\,C_{Lat}\,U^2 \tag{13}$$

A more detailed description of semi-numerical park load tool for fixed base VAWTs can be found in Sakib and Griffith (2022). The model also captures the tower drag. Tower drag is estimated using Equation 14.

$$F_{DTower} = 1/2\,\rho\,A_T\,C_D\,U^2 \tag{14}$$

where $A_T$ represents the frontal tower area, and $C_D$ is drag coefficient. For this analysis, a tapered cylindrical tower with an assumed $C_D$ of 1 was used. The measured thrust load for the locked platform case includes tower drag, while the lateral load does not account for it, as static tower drag force acts in the thrust load direction.

$$F_{ThM} = F_{DTower} + F_{Th} \tag{15}$$

The parked load estimating tool presented in Sakib and Griffith (2022) could only calculate the parked load for fixed base turbine. However, estimating park loads for floating offshore VAWTs is also important. The procedure of estimating park loads for floating offshore VAWTs is outlined below.

The parked loads for floating offshore VAWTs are directly correlated with the turbine dynamic motions. For example, updated sensor measurement thrust, and lateral loads are directly correlated with turbine pitch and roll motions, respectively. The detailed correlation of loads and turbine dynamic motions is shown in section 3.1.4.

The load cell is fixed on the tower base, please see Figure 1 for the load cell position. For the floating platform condition, the turbine tilts due to the coupled effects of wind load and floating system. Therefore, the load cell, which is fixed in the tower, also tilts the same as the turbine. Therefore, the load cell's measured loads will not be in the original thrust or lateral load direction. The measured park loads are in the tilted sensor measurement load direction. Readers are referred to Figure 6 to see the updated sensor measurement load direction. The figure only shows the updated sensor measurement thrust direction. The original thrust direction is in the windward direction. However, due to wind and wave loads the load sensor also tilts. Therefore, the measured data corresponds to the tilted updated sensor measurement direction. However, being consistent with the locked and floating platform terminologies, we will still call the floating parked loads as thrust and lateral loads.

[Figure]

**Figure 7.** Flowchart of parked load estimating semi-numerical tool.

The semi-numerical parked load measuring tool takes this tilting effect into consideration by including the resulting inertial (weight) effect due to tilting. Due to tilting, the load sensor measures a component of weight, along with the loads (thrust/lateral)

in the original direction. Figure 6 shows the concept of the weight effect present in the floating thrust load. In the test campaign, both the loads and turbine dynamic motions were recorded. We used the measured pitch and roll data to calculate the weight effect. One can use the aero-hydro floating VAWT model developed in Gao et al. (2022) to predict the tilt angles as pre-test data for input to the parked loads model presented here. The input aerodynamic loads can be supplied using CACTUS, while the hydrodynamic coefficients can be obtained from a commercial software like WAMIT [Lee and Newman (2006)] or an open source code like Capytaine [Ancellin and Dias (2019)]. Then the aerodynamic model and hydrodynamic code can be combined with a mooring model to predict the tilt angles as done in Gao et al. (2022).

$$Weight\,(due\,to\,pitch) = M\,g\,sin(\beta) \tag{16}$$

$$Weight\,(due\,to\,roll) = M\,g\,sin(\gamma) \tag{17}$$

Thrust and lateral loads including both aerodynamic loads and weight effects for the floating platform case are given in the mode as follows:

$$F_{ThFl} = F_{ThM} + M\,g\,sin(\beta) \tag{18}$$

$$F_{LatFl} = F_{Lat} + M\,g\,sin(\gamma) \tag{19}$$

where $\beta$ is measured pitch angle, $\gamma$ is measured roll angle, $M$ is turbine mass, and $g$ is the gravitational acceleration constant.

A flowchart of the semi-numerical parked load calculation process stated above is also shown in Figure 7. This figure contains park load modeling of both the locked base and floating base platforms. The floating base case compensates for the tilting effect taking gravitational effects into consideration.

The UTD's parked load tool for the fixed tower base case has already been validated in Sakib and Griffith (2022). The extension of the tool for floating tower base case is discussed and validated in the results and discussion section of this paper.

**3 Results and Discussion**

**3.1 Experimental data analysis**

The main objective of this study is to quantify and model the effects of number of blades, wind speeds, wind-wave-platform conditions, and rotor azimuth on parked loads. This section will show and discuss the effects of these conditions successively.

**3.1.1 Uncertainty in measurements**

We perform an uncertainty analysis for the experimental measurements as follows [Coquilla et al. (2007)]:

$$U = \sqrt{(\beta^2 + (t\sigma)^2)} \tag{20}$$

where, $U$ is resultant uncertainty, $\beta$ is bias uncertainty, $\sigma$ is standard deviation, and $t$ is the coverage factor of $T$ distribution. The value of $t$ is 1.96 at 95% confidence level. Bias uncertainty of the load cell and anomometer are 0.1%, and 2%, respectively.

[Figure]

**Figure 8.** Parked loads with error bars of 2B turbine for locked wind only condition

Error bars for experimentally measured parked loads of the 2B turbine for the locked wind only case are shown in Figure 8. It is observed that the thrust load exhibits an uncertainty of approximately 10 to 15%, whereas the lateral load shows slightly higher uncertainty. Uncertainty levels are consistent across all measurements; therefore, in the interest of clarity in results presentation, the uncertainty bars have been excluded from the other plots.

[Figure]

**Figure 9.** Effect of number of blades at locked tower base condition: a. thrust force b. lateral force. Note: horizontal bars shown are the variation in load due to azimuthal dependency, not uncertainty in the measurement.

**3.1.2   Effect of number of blades variation**

Both 2B and 3B floating VAWTs were tested in this test campaign. Variations in the number of blades, impacts the aerodynamic characteristics, structural dynamics, and overall efficiency of the turbine. For floating VAWTs, understanding the effect of the

number of blades on parked loads is essential to ensure stability and integrity under various operational and environmental conditions. This section explores the influence of the number of blades on parked loads, and their implications for floating VAWTs. Both 2B and 3B VAWTs were tested for fixed tower base and floating tower base conditions. However, the result is shown for the locked wind only case due to the fact that locked and floating platform conditions show a similar trend.

Figure 9 shows the parked loads as a function of wind speed for both the turbines. The horizontal bar in the figure shows variation due to azimuthal dependency, not uncertainty. It is observed 
[revised manuscript text omitted]
 2B turbine and 3B turbine, respectively. The experimental data represents the phase average of 5 revolutions, where the semi-numerical data is based on the last revolution values after the CACTUS simulation has converged. Figure 15a shows a comparison of non-dimensional forces, helping the evaluation of loads of different scale turbines, while Figure 15b shows a comparison of
335 dimensional case, which applies exclusively to this test turbine. Analysis shows generally a very good agreement between experiments and our parked loads model in terms of both magnitudes and rotor azimuth dependence. Both thrust and lateral

[Figure]

**Figure 14.** Experimental thrust and dynamic pitch motions in floating wind only condition for 2B turbine a. 2 ms[-1] b. 3 ms[-1] c. 4 ms[-1] d. 4.96 ms[-1].

loads amplitudes increase with the wind speed increment. The sharp decrease of experimental thrust load at 180° azimuthal location is due to the tower shadow effect, which is captured well in our model.

Both experimental and semi-numerical results show a clear relationship between wind speed and thrust force: higher wind speeds generate larger thrust forces. The periodic nature of the thrust force with respect to azimuth angle is consistent for the experiments and model.

The lateral force increases in magnitude with the increase wind speeds, keeping the mean force at almost zero. Experimental results show measurement noise due to sensor inaccuracies and inherent variability in the system.The results from semi-numerical model appear smoother compared to measured data. Again, the overall trends of force variation are captured similarly by both the experiment and the semi-numerical model.

Thrust force shows (Figures 15 and 16) strong resemblance between experiment and model, although the experimental results include measurement noise. Both model and experiment capture the cyclic nature (2 cycles for the 2B turbine and 3 cycles for the 3B turbine in a revolution) of the forces with respect to the azimuth angle.

[Figure]

(a) Non-dimensional loads

(b) Dimensional loads

**Figure 15.** Comparison between experimental and semi-numerical parked loads of 2B turbine for locked with wind only condition

**3.2.2 Floating wind only condition**

Modeling the floating case with wind comes with a new challenge. The challenge is to model the weight effect due to rotor tilting in the floating tower base case, as well as capturing the impact of tilting on aerodynamic loads. A detailed description of modeling parked loads at floating tower base condition has been shown in section 2.

This section is intended to analyze the modeled parked load data for floating wind only case in the absence of waves. Due to the floating condition, the rotor tilts under wind loading. Which adds a component of rotor weight in the measured updated sensor thrust and lateral direction. The rotor weight was already measured in the wind wave basin. The 2B and 3B turbines weigh 5.97 kg and 6.55 kg, respectively. This mass includes only the components mounted above the load cell, including the blades and tower mass. The measured mass excludes the mass of the lower tower section, load cell, generator and hull.

[Figure]

(a) Non-dimensional loads

(b) Dimensional loads

**Figure 16.** Comparison between experimental and semi-numerical parked loads of 3B turbine for locked with wind only condition

A representative weight effect in the thrust direction for the 2B turbine in the floating wind only condition is shown in Figure 17. This figure represents the pitch in red and turbine weight force in the measured thrust direction in blue. It is observed that, with the increase in wind speed, the weight effect also increases. It is a significant amount compared to the parked thrust load of similar wind speed. 2B turbine generates maximum thrust force of 5N at 4.96 ms$^{-1}$ for locked wind only case. Whereas, the weight effect for the same turbine at the floating condition due to tilting at 4.96 ms$^{-1}$ is 2.4 N, which is around 48% of the thrust load of the locked wind only case. Thus, including the rotor weight effect is very important while modeling the parked load for floating cases.

Figures 18 and 19 show the validation of UTD semi-numerical parked load with floating wind only condition for 2B and 3B turbines, respectively. The tool accurately predicts the trend for both thrust and lateral parked loads for the floating wind only case. However, the model slightly overpredicts the thrust load in some of the azimuthal locations, especially near 0°(360°) and 180° locations. Additionally, it underpredicts the lateral loads in certain azimuthal locations. The high frequency variation in

[Figure]

**Figure 17.** Weight effect in the thrust direction for 2B turbine in the floating with wind only tower base condition

loads, both in thrust and lateral parked loads, is due to turbine dynamics (roll/pitch) in the floating condition. The dimensional
370 and non-dimensional plots show similar trends.

**3.2.3 Floating tower base with wind and wave**

We now examine the third configuration of the test, which is floating with both wind and waves. The turbine weight effect is
also added with the fixed tower base parked load to calculate the parked load for wind and wave case. Regular wave with a
height of 0.155 m and a period of 1.61 s is used to measure the parked load at floating tower base with wind and wave condition.
375 And as the base is floating, it reflects a coupled effect due to wind, wave, and floating platform.

The validation in the case of floating with wind and wave case for the 3B turbine is shown in Figure 20. It is observed that the
tool cannot properly capture the dynamic nature of pitch and roll and respective dynamic thrust and lateral loads due to coupled

(a) Non-dimensional loads

(b) Dimensional loads

**Figure 18.** Comparison between experimental and semi-numerical parked loads of 2B turbine for floating with wind only condition

wind, wave, and floating platform effects, although it does handle the static pitch and roll motions. A prediction of pitch and roll motions requires integrating aerodynamic, wave, hydrodynamic, platform and mooring dynamic models. A code, such as WAMIT [Lee and Newman (2006)], or an open source code Capytaine [Ancellin and Dias (2019)], can be used to model the hydrodynamics, then coupled with mooring and aerodynamics models to predict the platform motion using an aero-hydro model [Gao et al. (2022)]. Such a model can be used for pre-test motion prediction. Here, the authors have restricted their study to the current semi-numerical parked load model and comparison with the experimental data. Overall, the semi-numerical tool well predicts the magnitude of parked loads, azimuthal dependence on loads, and the effects of wind speed, and solidity for all the wind-wave-platform conditions. Additionally, this tool accurately captures the tower shadow at the 180° azimuthal location.

[Figure]

(a) Non-dimensional loads

(b) Dimensional loads

**Figure 19.** Comparison between experimental and semi-numerical parked loads of 3B turbine for floating with wind only condition

**4    Conclusions**

Floating offshore VAWTs are showing promise for deep water offshore locations as they offer several advantages including a lower center of gravity, thus improving stability and reducing the risk of overturning. However, some aspects in the design of floating VAWTs must be studied, including parked loads, which are comparable in magnitude to operating loads [Sakib and Griffith (2022)], are thus critical design loads, yet no studies have measured parked loads under floating conditions.

This study experimentally investigates parked loads on floating VAWTs in a wind-wave basin, aiming to provide insights on the factors influencing parked loads. Additionally, this study seeks to gather data to improve and validate a semi-numerical parked load estimation tool for floating VAWTs under both wind-only and wind-and-wave conditions that is based on a vortex aerodynamics model of the rotor and an analytical model of tower drag. Further, this model effectively captures the effects of tower shadow, azimuth dependence and weight effects.

390

395

[Figure]

(a) Non-dimensional loads           (b) Dimensional loads

**Figure 20.** Comparison between experimental and semi-numerical parked loads of 3B turbine for floating with wind and wave condition

The study presents data on parked loads across varying wind speeds, solidity values, and wind-wave-platform conditions, examining the impact of gravitational loads from tilting and the correlation between tilt angles and parked loads. Validation of the semi-numerical estimation tool is also included.

The findings highlight the significance of wind speed, solidity, azimuth dependence, and wind-wave-platform conditions in determining parked loads for floating VAWTs. It was observed that VAWTs are subjected to substantial forces even in a stationary state, which necessitates robust structural designs to ensure their durability and safety.

The insights gained from this study underscore the importance of considering parked loads in the design phase of VAWTs. By incorporating validated parked loads models into the design process, engineers can develop more resilient turbines that can withstand the stresses encountered during non-operational periods. This is particularly crucial for enhancing the longevity and reliability of VAWTs, thereby making them a more viable option for renewable energy generation in diverse settings.The summary of the main findings of the paper can be stated as follows:

- Solidity (in terms of number of blades) influences the parked loads. The load variation (range) decreases as the number of blades increases.

- In floating conditions, average thrust load increases due to the gravitational load effect of turbine, while the amplitude of lateral load also increases.

- The floating wind-wave case exhibits more noisy loads due to dynamic nature compared to floating wind only case.

- The UTD's semi-numerical parked load tool quite accurately estimates the parked loads for both locked and floating platform conditions. It effectively captures the average load magnitude, load's azimuthal dependence, effects of wind speed, and number of blades. Tower shadow is also captured. However, this model is not formulated to predict the

415      dynamic nature of pitch and roll motions and respective dynamic thrust and lateral loads due to coupled wind, wave, and floating platform effects, although it effectively captures the average loads.

In conclusion, this study has advanced our understanding of the experimental parked loads on VAWTs and their impact on turbine performance. The results provide valuable guidelines for designing and implementing floating VAWTs that are both efficient and resilient. Future research should focus on further refining these findings through long-term field studies and

420    exploring innovative materials and design strategies to mitigate parked loads. By addressing these challenges, we can enhance the overall performance and adoption of VAWTs, contributing to the growth of sustainable wind energy solutions.

*Author contributions.* This work is performed during the PhD of MSH under the supervision of DTG as part of an Advanced Re- search Projects Agency–Energy (ARPA-E)-funded project named A Low-cost Floating Offshore Vertical Axis Wind System. MSH and DTG contributed to the analysis and interpretation of the data, and the manuscript was prepared by MSH with the help of DTG.

425    *Competing interests.* The contact author has declared that neither they nor their co-author has any competing interests.

*Acknowledgements.* The research presented herein was funded by the US Department of Energy Advanced Research Projects Agency-Energy (ARPA-E) under the ATLANTIS program with the project title "A Low-cost Floating Offshore Vertical Axis Wind System" associated with award no. DE-AR0001179. Any opinions, findings, and conclusions or recommendations expressed in this material are those of the authors and do not necessarily reflect the views of ARPA-E. The authors are grateful for the support of the ARPA-E program and staff, as

430    well as the project team.